# Revisiting Familiar Places in an Infinite World: Continuing RL in Unbounded State Spaces

## Abstract

Deep reinforcement learning (RL) algorithms have been successfully applied to train neural network control policies for many sequential decision-making tasks. However, prior work has shown that neural networks are poor extrapolators and deep RL algorithms perform poorly with weakly informative cost signals. In this paper we show that these challenges are particularly problematic in real-world settings in which the state-space is unbounded and learning must be done without regular episodic resets. For instance, in stochastic queueing problems, the state space and cost can be unbounded and the agent may have to learn online without the system ever being reset to states the agent has seen before. In such settings, we show that deep RL agents can diverge into unseen states from which they can never recover, especially in highly stochastic environments. Towards overcoming this divergence, we introduce a Lyapunov-inspired reward shaping approach that encourages the agent to first learn to be stable (i.e. to achieve bounded cost) and then to learn to be optimal. We theoretically show that our reward shaping technique reduces the rate of divergence of the agent and empirically find that it prevents it. We further combine our reward shaping approach with a weight annealing scheme that gradually introduces the pursuit of optimality and a log-transform of state inputs, and find that these techniques enable deep RL algorithms to learn performant policies when learning online in unbounded state space domains.

## 1 Introduction

Many deep reinforcement learning (RL) algorithms have been designed using environments (Fu et al., 2020; Brockman et al., 2016) that satisfy two criteria: (i) the state space and cost function are bounded, i.e., the agent will only experience a bounded range of numeric values; (ii) the environment is episodic and the agent will periodically be reset to some distribution of initial states. However, many real-world applications violate these criteria and instead have the following characteristics:

(i) The state space and range of inputs can be unbounded; and the incurred cost is unbounded.
(ii) Resets are difficult if not infeasible, requiring expensive manual or engineering efforts.

For example, in network load-balancing, one aims to minimize the number of jobs waiting in queues, and this number may grow arbitrarily large; in traffic intersection management, the number of vehicles may increase without bound.[1] In these examples, resetting the system (e.g., removing all vehicles at a traffic intersection) is costly or even inappropriate, particularly when a good simulator is absent and learning must be done in the real system (Sharma et al., 2021). Therefore, we need RL algorithms that are robust to potentially unbounded inputs and can learn quickly without the benefit of periodical resets. In this paper, we demonstrate that the failure of naïvely applying deep RL algorithms to a single, long episode when the state-space is unbounded. We then identify several key techniques which when combined, make deep RL algorithms able to solve these challenging problems.

**Challenges.** The unbounded state space means the agent must make predictions on states that are out of the range of previously seen states, which is known to be especially challenging for neural networks (Xu et al., 2021; Haley & Soloway, 1992; Barnard & Wessels, 1992; Chollet, 2017). Moreover, as we will see, it is common that the unbounded state space implies an unbounded cost function, which

---

[1]While the physics of the system may impose a finite limit on the state space (e.g., physical space at a road intersection), the limit is often so large that the problem is virtually unbounded and still foils existing methods.

can result in high variance updates and weakly informative learning signals (Arumugam et al., 2021). Both of these problems can severely hinder credit assignment. Meanwhile, the lack of episode resets reduces the chances that an RL agent can re-visit previously observed regions of the state-space and apply what was learned in earlier episodes to improve its estimation of the optimal actions. The unbounded state space and lack of resets are intricately linked. With poor credit assignment and poor extrapolation capabilities, the agent inevitably diverges into unvisited regions of the state space, and removing resets to previously visited states only worsens the divergence.

In this setting, we find that existing deep RL methods, as well as their natural variants, fail and cannot even keep the system *stable* with a bounded cost. That is, the agent *diverges*: it first encounters a state in which it has poor estimates of the optimal actions, it then makes a mistake, making it even more likely to encounter a new state in which it also has a poor estimate, and so on. The unboundedness of the state space means that there will always be *regions* of unvisited states, to which the agent must *extrapolate*. As the agent diverges into unvisited regions, it does not have the opportunity to refine estimation of the optimal actions in states seen so far. The lack of resets to previously visited states exacerbates the divergence. Finally, the unbounded cost results in high variance updates during policy learning, which further hinders credit assignment. Under this vicious cycle, the agent cannot even achieve a bounded cost, let alone the optimal cost. Towards tackling this divergence, our key insight is to encourage the agent to re-visit regions of the state space so that it can improve its estimation.

**Contributions.** Our contributions are three-fold and are summarized as follows:

1. We empirically show that existing deep RL methods diverge in continuing tasks with unbounded state spaces and high stochasticity. Common heuristics such as clipping, normalization and mapping the unbounded state space to a bounded range do not eliminate the divergence.

2. We introduce Stability Then OPtimality (STOP), an approach that prevents divergence and explicitly encourages the agent to re-visit regions of the state space and incur a bounded cost. The key idea of STOP is to enable the agent to refine its estimation of the optimal actions—even when the *current* policy is unstable and diverging—thus serving as a first step towards optimality. STOP is based on the combination of the following techniques: 1) a Lyapunov-inspired reward shaping approach that encourages the agent to be *stable*, i.e., achieve bounded cost (Shah et al., 2020), 2) a weight annealing scheme that prioritizes first learning stability and then gradually shifting towards optimality, and 3) state transformations that reduce the rate at which unbounded features can diverge and hence alleviate the extrapolation burden.

3. We conduct a thorough empirical study and analyze the role of different components of STOP on challenging real-world-inspired domains and show that STOP enables learning of highly performant RL policies.

See Section 6 for additional discussion of prior work on continuing RL and unbounded state spaces.

## 2 PRELIMINARIES

Consider an infinite-horizon Markov decision process (MDP) (Puterman, 2014), $\mathcal{M} := \langle \mathcal{S}, \mathcal{A}, \mathcal{P}, c, d_0 \rangle$, where $\mathcal{S} \subseteq \mathbb{R}^d$ is the state space, $\mathcal{A}$ the action space, $\mathcal{P} : \mathcal{S} \times \mathcal{A} \to \Delta(\mathcal{S})$ the transition dynamics, $c : \mathcal{S} \times \mathcal{A} \times \mathcal{S} \to \mathbb{R}_{\geq 0}$ the cost function, and $d_0 \in \Delta(\mathcal{S})$ the initial state distribution. Here we follow the control-theoretic convention and consider costs (the negation of rewards). Without loss of generality, we assume the cost is non-negative, and often restrict to cost-functions that are only dependent on the current state, $s_t$.

In the continuing task formulation, an agent, acting according to policy $\pi : \mathcal{S} \to \Delta(\mathcal{A})$, generates a *single* infinitely long trajectory: $s_0, a_0, s_1, c_0, a_1, s_2, ...$, where $s_0 \sim d_0$, $a_t \sim \pi(\cdot|s_t)$, $c_t = c(s_t, a_t, s_{t+1})$, and $s_{t+1} \sim \mathcal{P}(\cdot|s_t, a_t)$. Unlike typical settings in RL, there are no resets in this formulation. Accordingly, we consider the long-run average-cost objective (Naik et al., 2019; 2021):

$$J^O(\pi) := \lim_{T \to \infty} \frac{1}{T} \sum_{t=1}^{T} \mathbb{E}_\pi \left[ c(s_t, a_t, s_{t+1}) \right]. \tag{1}$$

The goal is to find an optimal policy that minimizes $J^O(\pi)$. Note that a necessary condition for optimality is stability, i.e., $J^O(\pi) < \infty$. Define the *differential* action-value function: $Q^\pi(s, a) := \lim_{T \to \infty} \mathbb{E}_\pi[\sum_{t=0}^{T}(c(s_t) - J^O(\pi))|s_0 = s, a_0 = a]$.

In this work, we make the following technical assumptions:

**Assumption 1** (Communicating MDP). *For any pair of states $s$ and $s'$, there exists a policy that can transition from $s$ to $s'$ in a finite number of steps with non-zero probability.*

**Assumption 2** (Bounded increment). *There exists a constant $B < \infty$ such that $\mathbb{E}_{s' \sim \mathcal{P}(\cdot|s,a)}|c(s') - c(s)| \leq B, \forall(s,a) \in \mathcal{S} \times \mathcal{A}$.*

**Assumption 3** (Norm equivalence). *The cost function satisfies $\underline{h}(\|s\|) \leq c(s,a,s') \leq \overline{h}(\|s\|), \forall(s,a,s')$ for some linear functions $\underline{h}$ and $\overline{h}$, where $\|\cdot\|$ is an arbitrary norm.*

Assumption 1 is standard for average-cost problems (Bertsekas, 2015); it ensures existence of a $\pi$ such that $J^O(\pi)$ is independent of the initial state. Assumption 2 is a mild regularity condition. Assumption 3 implies that stability/divergence in the state is equivalent to stability/divergence in the cost. These assumptions are satisfied in, e.g., queueing and network control problems where the state $s$ is the queue length vector, and one aims to minimize the time average of the total queue length $c(s) = \|s\|_1$, which is equivalent to minimizing *system latency* by Little's Law (Leon-Garcia, 2008).

## 3 CHALLENGES POSED BY UNBOUNDED STATE SPACES

In this section, we describe two core challenges that arise due to the unbounded state space: 1) poor credit assignment and 2) extrapolation burden.

**Challenge 1: Poor Credit Assignment.** A central challenge is that with an unbounded state space, directly optimizing the resulting unbounded cost function (Assumption 3) can lead to high variance updates during policy learning. Furthermore, the cost can be reduced only by a bounded amount $B$ per time-step (Assumption 2). When $B$ is small relative to the growing cost at the current state, any credit assigned to an action is weakly informative (Arumugam et al., 2021) and quickly overwhelmed by the diverging cost. Thus credit-assignment is hard, and this challenge cannot be eliminated by merely using a diferent state representation. Note that the above assumptions hold true in real-world settings such as queueing systems, where the cost function (the number of jobs in the system) may grow large while at most a finite number $B$ of jobs can enter or leave the system at each time-step. See Appendix C for more details.

**Challenge 2: Extrapolation Burden.** With an unbounded state space, there will always be *regions* of the state space unvisited by the agent. Note that an unbounded space is different from a continuous but bounded space: Suppose a neural network is trained on continuous state values from $[0, 1]$. If this bounded range is the entire state space, the neural network is typically making predictions for unseen values *inside* this region. On the other hand, in the unbounded state space setting, the neural network makes predictions for unseen values *outside* this region. It is well-known that while neural networks perform well in the former case, they often extrapolate poorly in the latter case (Xu et al., 2021; Haley & Soloway, 1992; Barnard & Wessels, 1992; Chollet, 2017), which results in the RL agent to act poorly. Furthermore, the poor extrapolation is exacerbated in the reset-free setting. Suppose there is a single initial state and the agent is reset to it every $k$ steps. Then agent's state visitation is effectively limited to a region of radius $k$ from the initial state. With enough training, the agent will improve its estimation of the optimal actions within this region and avoid divergence. However, if there are no resets, the agent will diverge into unvisited regions due to its poor extrapolation capabilities. One could potentially mitigate this burden by transforming the state space. However, as we will show, there are trade-offs between transformations.

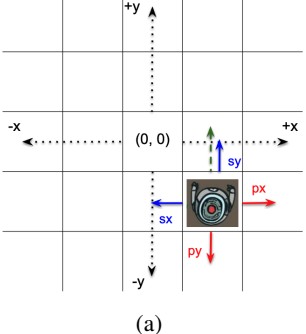

(a)

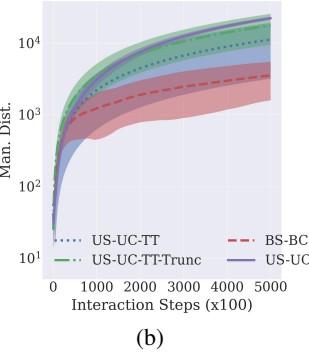

(b)

Figure 1: (a) Infinite gridworld environment. (b) Performance measured in Manhattan distance from origin vs. interaction steps. Results are over 5 trials and show the 95% confidence interval. Lower is better.

To illustrate the difficulty of learning in this setting, we consider an infinite gridworld (Figure 1(a), details in Appendix B) where the axes are unbounded in the positive and negative directions. The

agent tries to minimize time spent outside the origin, corresponding to a cost function being the Manhattan distance to the origin. The agent is pushed away from the origin and moves successfully towards the origin with certain probabilities. In Figure 1(b), we plot Manhattan distance (cost) vs. interaction time of various well-tuned average-reward PPO agents (Zhang & Ross, 2021) (see Table 1). In this case, increasing cost implies divergence as the agent is visiting newer states further way from origin. Despite the domain's simplicity, we find that commonly-used heuristics result in the agent to diverge. In particular, clippings (Engstrom et al., 2020), naïve transformations, resets, and state truncation all fail to stabilize, i.e., achieve a bounded cost, let alone the optimal cost.

| PPO Variant | Description | Reason for Divergence |
|---|---|---|
| US-UC | Optimize **U**nbounded **C**ost in **U**nbounded **S**tate space. Use state/reward clipping and normalizations. | Challenges 1 and 2 |
| BS-BC | Optimize **B**ounded **C**ost in **B**ounded **S**tate. $\tanh$ applied to state, cost $c$ is replaced with $-1/(c+1)$. | Challenge 1 and representation collapse by $\tanh$ |
| US-UC-TT | Trained with resets, then tested on reset-free setting. | Challenges 1 and 2 |
| US-UC-TT-TRUNC | Trained with resets and truncated state space $[-1000, 1000]^2$, then tested on unbounded state/cost and reset-free setting. | Challenges 1 and 2 |

Table 1: Common techniques to tackle the reset-free unbounded state and cost setting all diverge.

In the next section, we present an approach that tackles Challenges 1 and 2 through stability-based reward shaping, reverse annealing towards optimality and more sophisticated state transformations.

## 4    STOP: STABILITY THEN OPTIMALITY

We now introduce our method, Stability Then OPtimality (STOP), which is based on the insight of first encouraging stability (i.e., re-visiting regions of the state space and keeping the cost bounded) and then optimality (i.e., minimizing the cost). STOP consists of three algorithm-agnostic ingredients: 1) reward shaping for stability, 2) reverse-annealing the optimality cost, and 3) state transformations.

### 4.1    TACKLING CHALLENGE 1: STABILITY THROUGH REWARD SHAPING

As we described in Section 3, $J^O$ is difficult to minimize directly; even keeping $J^O$ bounded is challenging. To facilitate learning, we propose to use reward shaping to encourage the agent to be *stable* i.e., to not diverge and to incur bounded cost. A stable agent will be able to repeatedly revisit states and improve its estimates of the optimal actions. Once the agent is stable, it can start to optimize the original optimality objective $J^O$. To encourage stability, we propose using a *stability cost function* of the form $g(s_{t-1}, s_t) := \ell(s_t) - \ell(s_{t-1})$, where $\ell : \mathcal{S} \to \mathbb{R}_{\geq 0}$ is a *potential* or *Lyapunov* function satisfying $\liminf_{\|s\| \to \infty} \ell(s) = \infty$. Minimizing $g$ thus amounts to minimizing the *drift* of the Lyapunov function. In the following, we present a novel connection between this simple formulation and the notion of stability. Below we consider two instantiations of $\ell$.

A domain-independent choice for $\ell$ is to use the original cost function $c$, in which case $g(s_{t-1}, s_t) = c(s_t) - c(s_{t-1})$ is just the difference between consecutive costs. While straightforward, using $g$ has far-reaching consequences: 1) it induces a well-defined policy gradient even when the current policy is unstable, and 2) it provides immediate feedback on progress and encourages stability. We provide theoretical justifications for these properties in Section 4.1.1.

Another choice we consider is the quadratic potential function $\ell(s) = \|s\|_2^2$, where $\| \cdot \|_2$ is the Euclidean norm. In this case, we have $g(s_{t-1}, s_t) = \|s_t\|_2^2 - \|s_{t-1}\|_2^2 \approx 2\|s_{t-1}\|_2 \cdot (\|s_t\|_2 - \|s_{t-1}\|_2)$, which provides a reward signal if $\|s_t\|_2 < \|s_{t-1}\|_2$, especially when $\|s_{t-1}\|_2$ is large where it is crucial to prevent the state from further diverging. This choice often has better empirical performance. It is inspired by the literature on stochastic networks and control (Srikant & Ying, 2014; Neely, 2010), where a quadratic $\ell$ and its variants are popular candidates for a Lyapunov function.

Lyapunov functions are a standard tool in the *analysis* of stochastic systems, and it is known that a negative (expected) Lyapunov drift is a sufficient condition for stochastic stability (Meyn & Tweedie, 2012). Here we use it *algorithmically*. Note that we do not need to know an exact Lyapunov function—the above choices of $\ell$ are not exact in general. As shown in our experiments, an approximate Lyapunov function, even as crude as $\ell = c$, could still offer significant benefits.

### 4.1.1 THEORETICAL RESULTS

We investigate the theoretical properties of the stability cost function, focusing on the domain-independent choice $g(s, s') = c(s') - c(s)$. Define the corresponding average-cost objective:

$$J^{\mathrm{S}}(\pi) := \lim_{T \to \infty} \frac{1}{T} \sum_{t=1}^{T} \mathbb{E}_\pi g(s_{t-1}, s_t) = \lim_{T \to \infty} \frac{1}{T} \sum_{t=1}^{T} \mathbb{E}_\pi \left[ c(s_t) - c(s_{t-1}) \right]. \tag{2}$$

Later we will use $J^{\mathrm{S}}$ as a *regularizer* and combine it with the original optimality criterion $J^{\mathrm{O}}$.

Our first lemma shows that $J^{\mathrm{S}}(\pi)$ is always bounded and well-defined, regardless of whether $\pi$ is stable. This follows from the fact that $\mathbb{E}_\pi c(s_t)$ cannot grow faster than linearly under Assumption 2.

**Lemma 1.** *(Boundedness of g and $J^{\mathrm{S}}$) Under Assumptions 1 and 2, we have $\mathbb{E}_\pi [g(s_{t-1}, s_t)] \leq B, \forall t$ and $J^{\mathrm{S}}(\pi) \leq B$ for all policies $\pi$.*

We now define a new MDP, denoted by $\mathcal{M}^{\mathrm{S}}$, with state $x_t = (s_{t-1}, s_t)$ (i.e., a concatenation of two successive states) and instantaneous cost function $g(x_t) = c(s_t) - c(s_{t-1})$, in which case $J^{\mathrm{S}}(\pi)$ is the corresponding average-cost objective of $\mathcal{M}^{\mathrm{S}}$. Since $g$ and $J^{\mathrm{S}}$ are always bounded (Lemma 1), the action-value function, denoted by $Q^\pi_{\mathcal{M}^{\mathrm{S}}}$, is well defined for all policies $\pi$ under $\mathcal{M}^{\mathrm{S}}$. Using this connection, the proposition below characterizes the policy gradient of $J^{\mathrm{S}}$, allowing us to employ policy optimization methods to improve the current policy $\pi$ even when $\pi$ is not yet stable.

**Proposition 1.** *(Policy gradient of $J^{\mathrm{S}}$) Consider a policy class $\{\pi_\theta\}$ smoothly parameterized by $\theta$. Under Assumptions 1 and 2, for any $\theta$, the policy gradient is well-defined and admits the expression*

$$\frac{\mathrm{d}}{\mathrm{d}\theta} J^{\mathrm{S}}(\pi_\theta) = \mathbb{E}_\pi \left[ Q^{\pi_\theta}_{\mathcal{M}^{\mathrm{S}}} ((s_{t-1}, s_t), a_t) \nabla_\theta \log \pi_\theta(a_t | s_t) \right].$$

We say that the system is *rate-stable in mean* under the policy $\pi$ if $\lim_{T \to \infty} \frac{1}{T} \mathbb{E}_\pi \|s_T\| = 0$, that is, $\mathbb{E}_\pi \|s_T\| = o(T)$. Rate-stability (a.k.a. pathwise-stability) is a standard criterion in the queueing and stochastic network literature (Dai & Jennings, 2004; El-Taha & Stidham Jr, 2012). Our last proposition shows that minimizing $J^{\mathrm{S}}$ gives rate-stable policies. There are other formal notions of stability (Meyn & Tweedie, 2012; Shah et al., 2020); we leave it to future work to study the relationship between these alternative notions and our reward shaping approach.

**Proposition 2.** *($J^{\mathrm{S}}$ and rate stability) Under Assumptions 1–3, $\min_\pi J^{\mathrm{S}}(\pi) = 0$. Moreover, the system is rate-stable in mean under $\bar{\pi}$ if and only if $\bar{\pi} \in \arg\min_\pi J^{\mathrm{S}}(\pi)$.*

The proofs are deferred to Appendix A. While the above theoretical results focus on the choice $g(s', s) = c(s') - c(s)$, we find that other choices, such as the quadratic stability cost, also has good (and sometimes better) empirical performance.

### 4.2 APPLYING STABILITY + OPTIMALITY WITH REVERSE ANNEALING

While minimizing the shaped cost function $g$ yields a stable policy, we want an optimal policy. Thus, the agent must also pursue optimality, i.e., minimize the original objective $J^{\mathrm{O}}$. So that the agent first incurs bounded costs and then pursues optimal costs, we set the agent's objective at time-step $t$ to be a linear combination of $g(s_{t-1}, s_t)$ and $c(s_t)$:

$$\min_\pi \lim_{T \to \infty} \frac{1}{T} \sum_{t=1}^{T} \mathbb{E}_\pi \left[ \underbrace{g(s_{t-1}, s_t)}_{\text{stability cost}} + \lambda(t) \underbrace{c(s_t)}_{\text{optimality cost}} \right]. \tag{3}$$

The first part of the above objective is simply the stability objective $J^{\mathrm{S}}$. One may view $J^{\mathrm{S}}$ as a regularizer, which encourages stability throughout the interaction process. The coefficient $\lambda(t)$ in (3) is reverse annealed as a function of the time $t$ to increase emphasis on the optimality term over time. Specifically, we use the annealing schedule: $\lambda(t) = \tanh(\beta \max(t - \tau_{\mathrm{warmup}}, 0.01))$, where $\beta$ determines the speed at which $\lambda$ reaches 1 (its maximum value) and $\tau_{\mathrm{warmup}}$ determines at what time-step in the interaction process that $\lambda$ starts taking effect (MacGlashan et al., 2022).

By solving the optimization in Eq. (3) the agent first learns to be stable and then learns to be optimal. Since an optimal policy $\pi^*$ is by definition stable and satisfies $J^{\mathrm{S}}(\pi^*) = 0$,[2] the minimizer of the objective (3) will be the same as the optimal policy w.r.t. the original optimality criterion $J^{\mathrm{O}}$.

---

[2]The first part of Proposition 2 in fact holds for more general stability costs; see Remark 1 in Appendix A.

### 4.3 Tackling Challenge 2: State Transformations

The above two ingredients reduce the extrapolation burden (Challenge 2) by guiding the agent to re-visit regions of states with low costs. In domains with high stochasticity, the agent will still occasionally visit new states. To further mitigate the burden, we turn to different state transformations. These transformations are applied coordinate-wise to the state features only before inputting them into the policy and value networks; the cost function is computed based on the original state.

We consider the following transformation functions: 1) $\text{sigmoid}(x) := 1/(1 + e^{-x})$, 2) symmetric square root: $\text{symsqrt}(x) := \text{sign}(x)(\sqrt{|x| + 1} - 1)$ (Kapturowski et al., 2019), 3) symmetric natural log: $\text{symloge}(x) := \text{sign}(x) \ln(|x| + 1)$ (Hafner et al., 2023). All these functions 1) reduce divergence rate of the unbounded quantities and 2) preserve the ordering between them. From the agent's perspective, as the transformed states appear closer to one another, the extrapolation burden is further reduced. Of course, there are trade-offs between transformations. For example, while sigmoid reduces the extrapolation burden, it may effectively collapse the large states, making it challenging to differentiate between states/features. On the other hand, with symloge, the agent is still burdened by extrapolation but collapse is far less severe. For example, $\text{sigmoid}(100) \approx \text{sigmoid}(300) \approx 1$, $\text{symloge}(100) \approx 4.6$, $\text{symloge}(300) \approx 5.7$. See Appendix B for a visualization of these functions.

## 5 Empirical Study

We present an empirical study of STOP on real-world-inspired continuing tasks with unbounded state spaces, unbounded cost functions, and high stochasticity. We seek to answer the questions:

1. Does STOP improve the robustness of RL algorithms in this difficult setting?
2. How critical are the stability reward and state transformation components individually?
3. Is it better to optimize optimality from the beginning or gradually introduce optimality over time?

### 5.1 Setup

We first describe the environments, the algorithms we evaluate, and how we evaluate performance. We refer the reader to Appendix B for more details.

**Environments.** We conduct our experiments on the following environments.

**1. Two-dimensional goal-reaching infinite gridworld:** This domain was introduced in Section 3.

**2. Single-server allocation queuing:** At each time-step the server selects from a set of queues to serve. High stochasticity is due to job arrivals, failure in successfully serving a queue, and faulty connections. The optimality criterion is to minimize the average queue lengths. Note that in the faulty connection setting, finding the optimal policy is challenging even when the transition dynamics are known (Ganti et al., 2007). As we will see in the experiments, a smaller queuing setting with faulty connections can be significantly harder than a larger queueing setting with no faulty connections.

**3. Traffic control:** At each time-step a traffic controller must select a set of non-conflicting lanes to allow traffic where new cars arrive stochastically. We use the SUMO simulator (Behrisch et al., 2011; Alegre, 2019) and experiment with varying levels of traffic congestion on a given intersection design. The optimality criterion is to minimize the total waiting time across all cars. In this domain, SUMO models a real-life traffic situation by imposing a cap on the number of cars per lane. However, as we will show, existing deep RL algorithms still fail despite this cap.

**Algorithms.** Our baseline is average reward PPO (Zhang & Ross, 2021) since it is designed: 1) for the infinite horizon setting without discounting (Naik et al., 2021), 2) to be stable in the stochastic setting (Dai & Gluzman, 2022), and 3) to be robust to hyperparameter tuning. For the queueing environments, we also evaluate MaxWeight (see Appendix B), an algorithm with access to the transition dynamics.

**Online Evaluation and Training Procedure.** The agent starts from a random start state and a randomly-initialized policy and is never reset. We plot the true cost incurred by the agent vs. interaction time-steps. An increasing curve indicates divergence as the agent is visiting new states with increasing norm; a flat curve indicates that the agent is re-visiting states with bounded norms (stable); a decreasing curve indicates improvement towards optimality. See Appendix B for specifics.

## 5.2 MAIN RESULTS

We now compare vanilla PPO algorithm to PPO equipped with STOP in Figure 2. We hyperparameter sweep only over $\tau_{\text{warmup}}$ and $\beta$ and keep all other hyperparameters for STOP and the baseline the same (Raffin et al., 2021), and show results for agents that achieved the lowest true optimality cost at the end of interaction time. We evaluate both stability variations: 1) $\ell(s) = c(s) = \|s\|_1$ (linear) and 2) $\ell(s) = \|s\|_2^2$ (quadratic). All STOP agents use the symlog transform.

In all experiments, the naïve PPO agents, which optimized the optimality criterion directly and used no state transformations, diverged. On the other hand, STOP agents are able to achieve stability. In the highly stochastic queueing environments only, we found that using the linear stability cost and optimality cost is insufficient: the linear stability cost (bounded between $[-1, 1]$) and optimality cost have different magnitude scales, which can cause the latter to dilute the former, effectively eliminating any benefit of the stability cost. In these cases, we replaced the optimality cost $c(s_t)$ with $-1/(c(s_t) + 1)$.

While in some cases the linear stability cost agent performed marginally better than the quadratic cost agent, the latter has two advantages in the evaluated environments: 1) it is on a similar scale as the optimality cost $c$, mitigating any dilution, and 2) it allows us to keep and optimize the true optimality cost; in comparison, using the linear cost may require taking the reciprocal of $c$, which is not equivalent to the true optimality criterion.

While optimizing *only* the stability criterion led to low true cost, including the optimality criterion lowered it further. Refer to Figure 6 in Appendix B for these results.

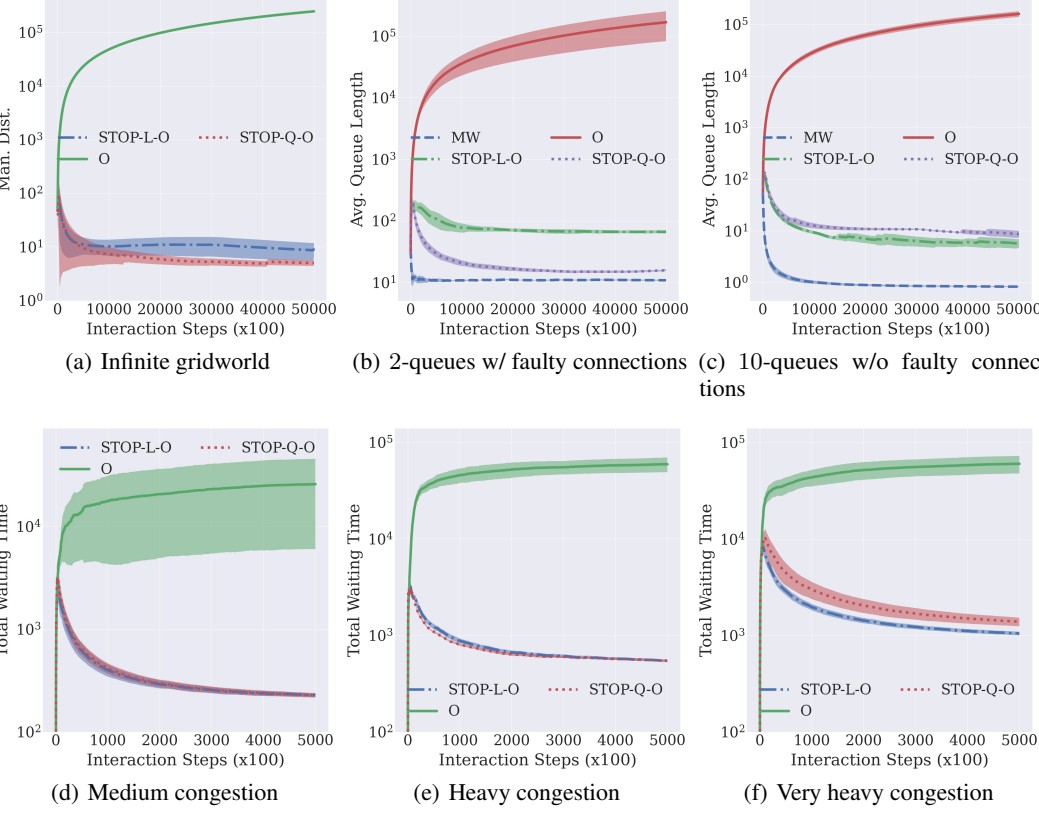

(a) Infinite gridworld     (b) 2-queues w/ faulty connections     (c) 10-queues w/o faulty connections

(d) Medium congestion     (e) Heavy congestion     (f) Very heavy congestion

Figure 2: True optimality criterion vs. interaction time-steps on infinite gridworld, two queue networks variants, and traffic control environment. Lower is better. Algorithms are PPO (O) vs. STOP + PPO which use all STOP components, where we evaluate the linear (STOP-L-O) and quadratic (STOP-Q-O) stability cost function. For the queueing environment, we also report the performance of MaxWeight (MW). Recall that unlike MW, STOP does not know the transition dyanmics. Performance metrics are computed over 5 trials with 95% confidence intervals. The vertical axis is log-scaled.

## 5.3 ABLATION STUDIES

We have showed that STOP enables learning in highly stochastic environments with unbounded state spaces and cost functions. We now analyze the importance of different components of STOP.

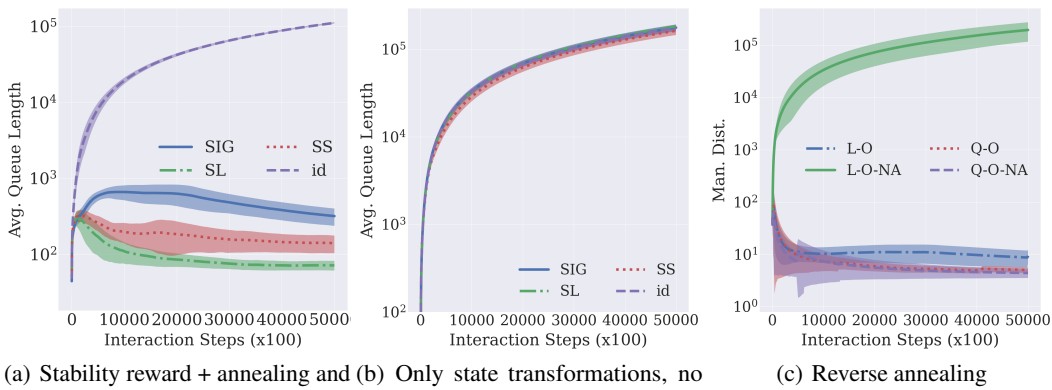

(a) Stability reward + annealing and varying state transformations

(b) Only state transformations, no stability reward

(c) Reverse annealing

Figure 3: Ablations. True optimality criterion vs. interaction time-steps on (a) 2 queue setup, (b) 10 queue setup, and (c) infinite gridworld. Lower is better. Performance metrics are computed over 5 trials with 95% confidence intervals. The vertical axis is log-scaled. id: identity; SIG: sigmoid, SS: symsqrt; SL: symloge.

### 5.3.1 STABILITY REWARD + ANNEALING WITH VARYING STATE TRANSFORMATIONS

In this experiment, we seek to understand the importance of the state transformation component of STOP. We evaluate the performance of a STOP PPO agent with $g(s_{t-1}, s_t) := c(s_t) - c(s_{t-1})$, different state transformations from Section 4.3, and reverse annealing optimality applied to the unbounded cost. From Figure 3(a), applying state transformations result in significantly better performance compared to no application of such a transformation. The significant improvement over no state transformations (id) suggests that state transformations are indeed critical and that in very difficult environments the stability cost is insufficient. However, we also observe that balancing between squashing values too closely to each other (e.g. sigmoid) vs. keeping them reasonably far apart (e.g. symloge) is important. If values are extremely squashed together, it may be difficult for the agent to distinguish between states and features of a state. If they are too far apart, the extrapolation burden may be less mitigated.

### 5.3.2 STATE TRANSFORMATIONS WITH NO STABILITY REWARD

In Figure 3(b), we remove the stability cost. We observe that applying only state transformations is insufficient for the agent to stablize and the stability reward component of STOP is critical.

### 5.3.3 STOP WITH AND WITHOUT REVERSE ANNEALING OPTIMALITY

We study whether it is better to gradually optimize optimality (apply reverse annealing) or to always optimize optimality (setting $\lambda(t) = 1$). In Figure 3(c), we show the performance of different STOP agents using the linear (L-O) and quadratic (Q-O) stability cost and symloge state transform when using reverse annealing vs. not using reverse annealing (-NA). The linear STOP agent with no reverse annealing performed poorly. We attribute this bad performance to the fact that initially the policy is unstable, causing the agent to incur an unbounded optimality cost, which in turn dilutes any benefit of the stability cost, so the agent never learns to be stable. On the other hand, we found the quadratic STOP agent to be more robust. This robustness is due to the fact that the quadratic stability cost is on a similar magnitude scale to the optimality cost, thus both criteria have similar influence. Thus, it is preferable to optimize optimality gradually.

In terms of how to introduce optimality, we found that it was better to introduce optimality later (larger $\tau_{\text{warmup}}$) and slowly (smaller $\beta$) during interaction (see Appendix B). That is, it is more important that the agent first learn to stabilize and then pursue optimality than pursuing optimality from the start.

## 6 RELATED WORK

**Continuing RL.** Our work focuses on continuing RL tasks in which interaction never terminates and performance is measured online (Sutton & Barto, 2018). This setting has been described with the autonomous RL (Sharma et al., 2021) or single-life RL (Chen et al., 2022) formalisms. In many practical set-ups, it is infeasible to reset the agent to a new initial state. Recent work has thus considered *reset-free* RL where the agent learns to reset itself (Eysenbach et al., 2018; Zhu et al., 2020; Gupta et al., 2021; Han et al., 2015). However, these works still require a manual reset (Eysenbach et al., 2018) or use policies learned on one task as a reset policy in another task (Gupta et al., 2021). Our work performs no resets and evaluates performance on a single infinitely-long task.

In continuing RL, the natural performance measure of an agent is average-reward (Sutton & Barto, 2018; Naik et al., 2019). Our work is different from prior average-reward RL work (Mahadevan, 1996; Schwartz, 1993; Wan et al., 2021; Wei et al., 2020; Zhang & Ross, 2021; Zhang et al., 2021) in that we tackle the challenge of unbounded state spaces and cost functions.

To avoid possible confusion, we emphasize that our work focuses on *continuing* RL and not *continual* RL. The latter tends to address non-stationarity in the transition dynamics and reward function, and much work in continual RL still uses an episodic task formulation. Our setting, on the other hand, assumes that the agent is solving *one* task with stationary dynamics in a *single* episode.

**Stability and Unbounded State Spaces in RL.** The concept of stability in RL has largely been overlooked. Stability is related to the notion of *safety* in RL (García & Fernández, 2015), but with crucial differences. Safety is typically defined as hard constraints on individual states and/or actions (Hans et al., 2008; Dalal et al., 2018; Dean et al., 2019; Koller et al., 2018), or soft constraints on the expected costs over the trajectory (Achiam et al., 2017; Chow et al., 2018; Yu et al., 2019). In contrast, stability concerns the long-run asymptotic behavior of the system, which cannot be immediately written as constraints over the state/action or budget on some cost.

Some work considers control-theoretic notions of stability (Vinogradska et al., 2016; Berkenkamp et al., 2017). While related, these results mostly consider systems with deterministic and partially unknown dynamics. For example, Westenbroek et al. (2022) propose a cost-shaping approach with similarities to ours, but focusing on deterministic dynamics and hence a different notion of stability; they consider discounted costs, and do not study the continuing setting with unbounded states.

The work of Shah et al. (2020) considered stochastic stability for RL in the unbounded state space setting. However, their work assumed a tabular setting, relied on access to the model of the environment, and ignored optimality and focused exclusively on stability. Our work makes stability practical for deep RL without having access to the environment model, and combines stability with optimality. In the few other works that do consider stability (Dai & Gluzman, 2022; Liu et al., 2022), they assume that a stable policy is given and use it as a starting point for learning an optimal policy. We make no such assumption and try to learn a stable policy directly from a random policy.

The combination of unbounded space and continuing setting is scarce in the RL literature. Existing works acknowledge the challenges from unboundedness, but they either artificially bound the state space (Liu et al., 2022), or assume the episodic training setting (Dai & Gluzman, 2022). In Section 3, we showed that these adjustments did not solve the reset-free unbounded state space setting.

## 7 CONCLUSION

We introduced STOP that prevents divergence of deep RL agents in highly stochastic, reset-free and unbounded state space and cost settings. STOP guides the agent to re-visit regions of the state space and achieve stability, and to then pursue optimality. This general principle can be useful for many RL settings and RL algorithms.

We highlight some limitations and future directions. STOP relies on the true optimality cost to be dense. It would be interesting to develop stability techniques with sparse rewards. Also, while STOP substantially improved RL-based baselines, our comparison to a non-RL baseline in the queueing environments suggests that further improvement is possible. This gap could potentially be closed by examining the loss of plasticity (Nikishin et al., 2022). Another interesting direction is to combine STOP with sophisticated off-policy algorithms such as RAINBOW (Hessel et al., 2017).

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

## A  THEORETICAL RESULTS

In this section we provide the proofs for our theoretical results, which are restated below for readers' convenience. The starting point of our proofs is to rewrite the stability objective $J^S$ as follows:

$$\begin{aligned}
J^S(\pi) &= \lim_{T\to\infty} \frac{1}{T} \sum_{t=1}^{T} \left( \mathbb{E}_\pi \ell(s_t) - \mathbb{E}_\pi \ell(s_{t-1}) \right) \\
&= \lim_{T\to\infty} \frac{1}{T} \left( \mathbb{E}_\pi \ell(s_T) - \mathbb{E}_\pi \ell(s_0) \right) \\
&\overset{(i)}{=} \lim_{T\to\infty} \frac{1}{T} \mathbb{E}_\pi \ell(s_T) \\
&= \lim_{T\to\infty} \frac{1}{T} \mathbb{E}_\pi c(s_T),
\end{aligned} \tag{4}$$

where the equality (i) holds since $\mathbb{E}_\pi c(s_0)$ is independent of $T$.

**Lemma 1.** *(Boundedness of g and $J^S$) Under Assumptions 1 and 2, we have $\mathbb{E}_\pi[g(s_{t-1}, s_t)] \le B, \forall t$ and $J^S(\pi) \le B$ for all policies $\pi$.*

*Proof.* To prove boundedness of $g$, we observe that for all $t$:

$$\begin{aligned}
\mathbb{E}_\pi |g(s_t, s_{t+1})| &= \mathbb{E}_\pi |c(s_{t+1}) - c(s_t)| \\
&\overset{(i)}{=} \mathbb{E}_{(s_t, a_t)\sim\pi} \mathbb{E}_{s_{t+1}\sim\mathcal{P}(\cdot|s_t, a_t)} |c(s_{t+1}) - c(s_t)| \\
&\overset{(ii)}{\le} \mathbb{E}_{(s_t, a_t)\sim\pi} B \\
&= B,
\end{aligned}$$

where in step (i) we use the law of iterated expectation and denote by $\mathbb{E}_{s_t\sim\pi}$ the expectation w.r.t. $(s_t, a_t)$ generated by following the policy $\pi$, and in step (ii) we use Assumption 2.

It follows that

$$\begin{aligned}
\mathbb{E}_\pi c(s_T) &= c(s_0) + \sum_{t=0}^{T-1} \mathbb{E}_\pi [c(s_{t+1}) - c(s_t)] \\
&\le c(s_0) + \sum_{t=0}^{T-1} \mathbb{E}_\pi |g(s_t, s_{t+1})| \\
&\le c(s_0) + BT,
\end{aligned}$$

whence

$$\begin{aligned}
J^S(\pi) &= \lim_{T\to\infty} \frac{1}{T} \mathbb{E}_{\pi, s_0} c(s_T) \\
&\le \lim_{T\to\infty} \frac{1}{T} (c(s_0) + BT) \le B.
\end{aligned}$$

$\square$

For the next result, we recall that for the average-cost MDP $\langle \mathcal{S}, \mathcal{A}, \mathcal{P}, c, d_0 \rangle$ and a policy $\pi$, the *differential* action-value function is defined as $Q^\pi(s, a) := \lim_{T\to\infty} \mathbb{E}_\pi[\sum_{t=0}^{T}(c(s_t) - J^O(\pi))|s_0 = s, a_0 = a]$. Also recall that $Q^\pi_{\mathcal{M}^S}$ is the differential action-value function for the MDP $\mathcal{M}^S$ associated with the stability cost $g$.

**Proposition 1.** *(Policy gradient of $J^S$) Consider a policy class $\{\pi_\theta\}$ smoothly parameterized by $\theta$. Under Assumptions 1 and 2, for any $\theta$, the policy gradient is well-defined and admits the expression*

$$\frac{d}{d\theta} J^S(\pi_\theta) = \mathbb{E}_\pi \left[ Q^{\pi_\theta}_{\mathcal{M}^S}((s_{t-1}, s_t), a_t) \nabla_\theta \log \pi_\theta(a_t|s_t) \right].$$

*Proof.* The claim follows from applying the average-cost Policy Gradient Theorem (Sutton et al., 1999, Theorem 1) to the MDP $\mathcal{M}^S$ with $g$ as the cost function, and using the well-known identity $\pi_\theta(a|s) \cdot \nabla_\theta \log \pi_\theta(a|s) = \nabla_\theta \pi_\theta(a|s)$. $\square$

**Proposition 2.** *($J^{\text{S}}$ and rate stability) Under Assumptions 1–3, $\min_\pi J^{\text{S}}(\pi) = 0$. Moreover, the system is rate-stable in mean under $\bar{\pi}$ if and only if $\bar{\pi} \in \arg\min_\pi J^{\text{S}}(\pi)$.*

*Proof.* By Assumption 1, there exists a stable policy $\pi_0$ such that $\mathbb{E}_{\pi_0} c(s_t) \leq C, \forall t$ for some constant $C < \infty$. Therefore,

$$0 \leq \min_\pi J^{\text{S}}(\pi) \leq J^{\text{S}}(\pi_0) = \lim_{T\to\infty} \frac{1}{T}\mathbb{E}_{\pi_0} c(s_T) \leq \lim_{T\to\infty} \frac{1}{T} \cdot C = 0, \tag{5}$$

hence $\min_\pi J^{\text{S}}(\pi) = J^{\text{S}}(\pi_0) = 0$.

Consequently, we have $\bar{\pi} \in \arg\min_\pi J^{\text{S}}(\pi)$ if and only if $J^{\text{S}}(\bar{\pi}) = 0$. This is in turn equivalent to

$$0 = J^{\text{S}}(\bar{\pi}) = \lim_{T\to\infty} \frac{1}{T}\mathbb{E}_\pi c(s_T) = \lim_{T\to\infty} \frac{1}{T}\mathbb{E}_\pi c(s_T),$$

which is the definition of rate stability in mean. $\qquad\square$

**Remark 1** (Generalization of Proposition 2). *Inspecting equation (5) above, we see that the conclusion $J^{\text{S}}(\pi_0) = 0$ in fact holds for other stability costs $g(s_t, s_{t+1}) = \ell(s_{t+1}) - \ell(s_t)$ beyond $\ell = c$. In particular, suppose the stable policy $\pi_0$ and the Lyapunov function $\ell$ satisfy $\mathbb{E}_{\pi_0}\ell(s_t) \leq C < \infty, \forall t$. Note that this property holds when, for example, we use a quadratic Lyapunov function $\ell(s) = \|s\|_2^2$ and the Markov chain $(s_t)_{t\geq0}$ induced by $\pi_0$ has a limit distribution with a finite second moment. Many queueing and stochastic control problems have this property, since the limit distribution is typically sub-exponential (Hajek, 1982). In this case, similarly to equation (5), we have*

$$0 \leq J^{\text{S}}(\pi_0) \overset{(i)}{=} \lim_{T\to\infty} \frac{1}{T}\mathbb{E}_{\pi_0}\ell(s_T) \leq \lim_{T\to\infty} \frac{1}{T} \cdot C = 0, \tag{6}$$

*where step (i) follows from equation (4). It follows that $J^{\text{S}}(\pi_0) = 0$.*

## B  SUPPORTING CONTENT AND EMPIRICAL RESULTS

In this section, we include additional details and experiments that complement the main results. We also include the code in the supplementary zip file.

### B.1  VISUALIZATIONS OF STATE TRANSFORMATIONS

To provide better intuition of the different state transformations we considered in Section 4.3, we visualize them in Figure 4.

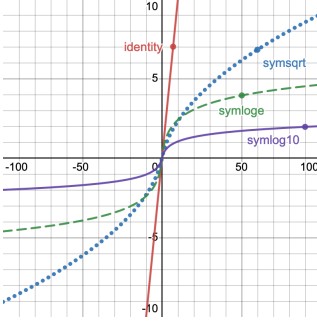

Figure 4: Visualizations of transformation functions.

### B.2  ENVIRONMENTS

In this section, we provide additional details about the server allocation and traffic control environment.

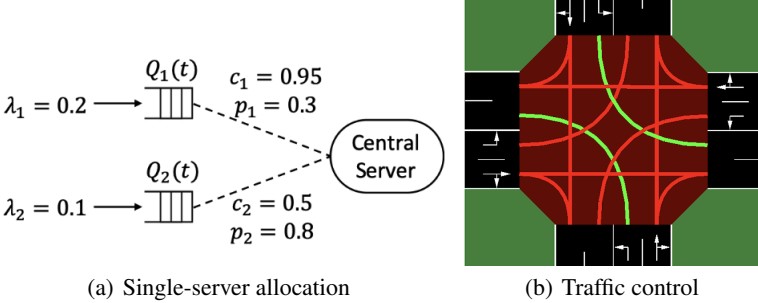

(a) Single-server allocation          (b) Traffic control

Figure 5: Left: Server-allocation. Image taken from Liu et al. (2019). Right: An example intersection of the traffic control environment. Image taken from Alegre (2019).

**Gridworld** In this domain, an agent is randomly spawned within some region of the grid and must try to move towards the origin by selecting between two actions: moving vertically or horizontally towards the goal. The state of the agent is the unbounded 2D coordinate location of the agent. With probabilities $p_x = 0.2, p_y = 0.15$ the agent is pushed away from the origin along the respective axis. After an agent selects an axis to move along, its movement along that axis will be successful with probability $s_x = 0.3, s_y = 0.8$ for the respective axis; if it fails to move, it will stay in the same location unless pushed away. The cost function is the Manhattan distance between the agent and the origin.

**Single-server allocation queueing** In this environment, there is a single central server that must select among a set of queues to serve. In general, there can be up to $N$ queues (Figure 5 show a sample 2 queue setup). At each time-step, new jobs arrive in each queue following a Bernoulli process with probability $\lambda_i$ for queue $i$. Note that at each time-step, at most one job enters each queue, which means more than one job may enter the whole system in total. At each time-step, the server must select among the $N$ queues to serve. A successfully served queue will mean a job will exit that queue, so at most one job can exit the system at a given time-step. When a server selects a queue $i$, the server succeeds in making a job exit only if: it can connect to queue $i$ (which is dependent on the connectivity probability, $c_i$) and the job is successfully served (which is dependent on the service probability, $p_i$). The state of the server is queue length of each queue and $0/1$ flag indicating whether the server can connect to a specific queue, resulting in $2N$-dimensional state. The action space is index of the queue, resulting in $N$ dimensions. The goal is to minimize the average queue length. In the non-faulty connection setting, all the connectivity flags are $1$. The optimal policy in the faulty connection setting is an open problem (Ganti et al., 2007).

The Bernoulli probability parameters of the tested environments are:

1. 2-queue with faulty connections
   - Arrival rates: $\lambda_1 = 0.2, \lambda_2 = 0.1$
   - Service rates: $p_1 = 0.3, p_2 = 0.8$
   - Connection probabilities: $c_1 = 0.7, c_2 = 0.5$

2. 10-queue with non-faulty connections
   - Arrival rates: $\lambda_1 = 0.05, \lambda_2 = 0.01, \lambda_3 = 0.2, \lambda_4 = 0.4, \lambda_5 = 0.05, \lambda_6 = 0.01, \lambda_7 = 0.02, \lambda_8 = 0.01, \lambda_9 = 0.015, \lambda_{10} = 0.01$
   - Service rates: $p_1 = 0.9, p_2 = 0.85, p_3 = 0.95, p_4 = 0.75, p_5 = 0.9, p_6 = 0.9, p_7 = 0.85, p_8 = 0.9, p_9 = 0.9, p_{10} = 0.85$
   - Connection probabilities: $c_i = 1$ for all $1 \leq i \leq 10$

When evaluating the RL algorithms, we compare their performance to MaxWeight (Tassiulas & Ephremides, 1990; Stolyar, 2004), a well-known algorithm that achieves stability for a certain class of queueing scenarios, but which relies on the knowledge of the system model (i.e. some parts of the transition dynamics) and it is generally unknown how far MaxWeight is from optimality. It is a very strong non-RL baseline from decades of research from the stochastic networking community.

**Traffic control**    In this environment, a traffic controller must select from a set of phases (shown in green in Figure 5), a set of non-conflicting lanes, to allow cars to move. At each time-step, new cars arrive in each lane at different rates, which determines the traffic congestion level. In our experiments, we considered medium to very high levels of traffic congestion. The state is the number of cars waiting in each lane along with indicator flags for which lanes have a green and yellow light. The action space is the number of phases. The state space is 21 dimensions and the action space is 4. The goal is to minimize the total waiting time of all the cars. To model a real-life traffic situation, the SUMO simulator places a cap of $\approx 100$ on each lane. We use the SUMO simulator implementation (Behrisch et al., 2011; Alegre, 2019).

For exact traffic demands used in the experiments, see the `sumo/nets/big-intersection/generator.py` file in the attached code.

### B.3    ADDITIONAL EMPIRICAL SETUP DETAILS

**PPO Training**    We train average-reward PPO (Zhang & Ross, 2021; Dai & Gluzman, 2022) using the default hyperparameters (network architecture, learning rate, mini batches, epochs over the dataset etc.) in the stablebaselines code base (Raffin et al., 2021). For all algorithms and variations, we fix the time interval between policy updates during the interaction to be 512 time-steps. For STOP, we hyperparameter sweep only over $\beta = \{1e^{-4}, 1e^{-5}, 1e^{-6}\}$ and $\tau_{\text{warmup}} = \{10^6, 2 \cdot 10^6, 3 \cdot 10^6\}$. Whenever we perform a hyperparameter sweep, we select the agent that achieves the lowest performance metric value at the end of the experiment (Manhattan distance for gridworld, average queue length for queueing, and total waiting time for traffic control).

### B.4    OPTIMIZING ONLY STABILITY

We also find that STOP agents that optimize *only* the stability criterion was sufficient to stabilize the agent, but including the optimality criterion led to better performance (Figure 6). Note that these STOP agents use the symloge state transform.

### B.5    VARYING REVERSE ANNEALING SCHEDULE WITH $\beta$ AND $\tau_{\text{WARMUP}}$

In Section 5, we showed the impact of gradually introducing optimality vs. optimizing optimality from the beginning of the interaction process. In these set of experiments, we analyze the impact of the reverse annealing schedule: varying $\tau_{\text{warmup}}$ (when optimality starts to get introduced) and $\beta$ (how fast it gets introduced). We show results on the infinite gridworld domain for both the linear and quadratic stability cost in Figure 7. Note on the legend: T# means $\tau_{\text{warmup}} = \# \cdot 10^6$ and B# means $\beta = 10^{-\#}$.

We find that in general: slowly introducing optimality (smaller $\beta$) and introducing optimality later on in the interaction process (larger $\tau_{\text{warmup}}$) is better for achieving lower true cost. This finding aligns with our conclusion that it is better for the agent to first learn how to be stable before trying to be optimal.

### B.6    HARDWARE FOR EXPERIMENTS

For all experiments, we used the following compute infrastructure:

- Distributed cluster on HTCondor framework
- Intel(R) Xeon(R) CPU E5-2470 0 @ 2.30GHz
- RAM: 5GB
- Disk space: 5GB

## C    POOR CREDIT ASSIGNMENT WITH UNBOUNDED COSTS

As noted in Section 3, the unbounded cost can make credit assignment challenging. In this section, we make this intuition more concrete with an example. Due to Assumption 3, the cost function can

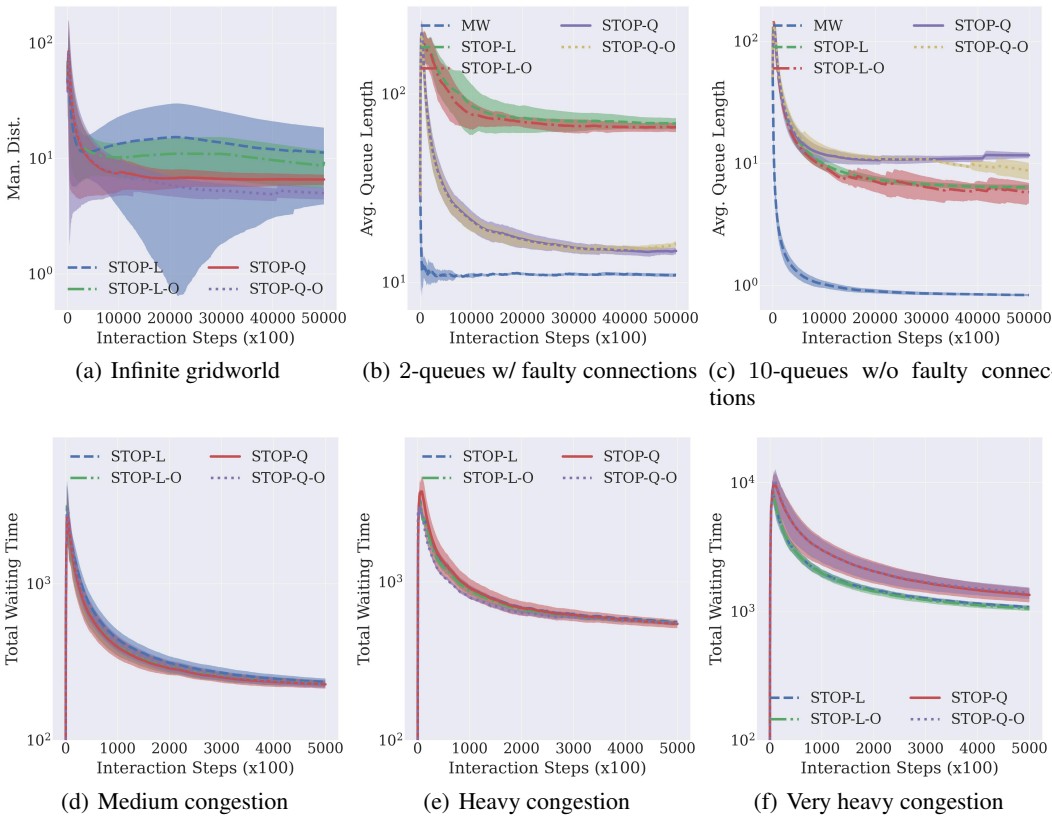

Figure 6: True optimality criterion vs. interaction time-steps on infinite gridworld, two queue networks variants, and traffic control environment. Lower is better. Algorithms are STOP with and without optimality introduced, where we evaluate the linear (L) and quadratic (Q) stability cost function. For the queueing environment, we also report the performance of MaxWeight (MW). Performance metrics are computed over 5 trials with 95% confidence intervals. The vertical axis is log-scaled.

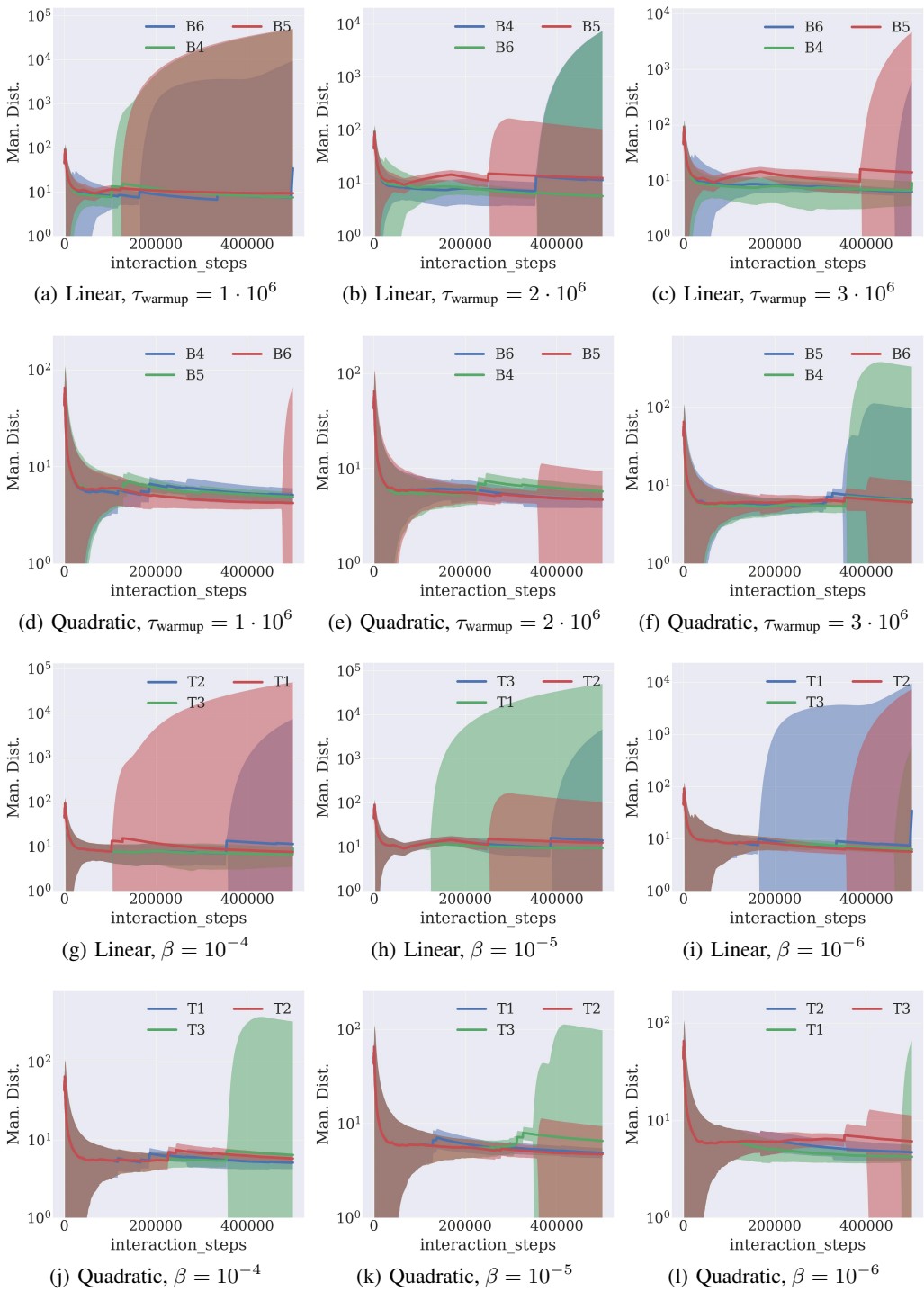

Figure 7: Manhattan distance vs. interaction time-steps on the infinite gridworld for different values of $\beta$ and $\tau_{\text{warmup}}$ for the linear and quadratic stability cost functions. Performance metrics are computed over 5 trials with 95% confidence intervals. Lower is better. T# means $\tau_{\text{warmup}} = \# \cdot 10^6$ and B# means $\beta = 10^{-\#}$

be unbounded. Furthermore, Assumption 2 implies that the cost can be reduced only by a bounded amount $B$ per time-step. As mentioned, when $B$ is small relative to the unbounded cost at the current state, then any credit assigned to an action is weakly informative (Arumugam et al., 2021) and thus credit-assignment is hard.

To understand this challenge, consider a queueing example where the true cost function is the average queue length, and we have 2 queues where in each time step, at most 1 job can leave and at most 2 new jobs can arrive into the system. Say the state is $s = (q_1, q_2)$ where $q_i$ is the length of the $i$th queue. Let $s = (50, 100)$ and consider two scenarios: i) the agent chooses $q_1$ and successfully reduces its length; ii) the agent chooses $q_2$ and fails to reduce its length. In both situations, a new job arrives into each queue. The differential advantage function of $q_1$ and $q_2$ are $A([50, 100], q_1) = -75.5 + V([50, 101]) - V([50, 100]) - \rho$ and $A([50, 100], q_2) = -76 + V([51, 101]) - V([50, 100]) - \rho$ respectively, where $\rho$ is the average reward of the policy. We can see that choosing $q_1$ was the better action but the credit assigned to it is marginally better than credit assigned to the action that chose $q_2$. To properly assign credit, the agent has to encounter a large number of these samples. However, as we have discussed, the divergence prevents the agent from refining its estimation of the optimal actions. In the first scenario, $B = 0.5$ and in the second, $B = 1$, which are both much smaller than the instaneous cost $\|s\|_1 \approx 75$.

## D    STOP ENCOURAGES AGENT TO REVISIT STATES

In this section, we show that STOP does encourage the agent to re-visit states. In Figure 8, we plot a 15 by 15 heatmap of the sub-grid of the infinite gridworld environment where the center of the grid is the origin. Each box corresponds to the original agent's state i.e. the native coordinate location of the agent. In this plot, the color of an individual box denotes the fraction of time that the agent visits the given state. This fraction is calculated as a probability based on the state-visitation frequencies from time 0 up till a given time $T$. White corresponds to low and and dark blue corresponds to high visitation.

We can see that initially the STOP agent spends almost no time inside this 15 by 15 grid. However, over time the fraction of time that the agent spends within the sub-grid increases (from 0.09 to 0.72; see horizontal axis label: fraction of time spent within sub-grid). That is, over time STOP encourages the agents to visit the states closer to the origin.

## E    STATE TRANSFORMATIONS ENCOURAGE AGENT TO REVISIT STATES

In this section, we show that state transformations encourage revisiting states *from the agent's perspective*. Similar to Figure 8, we plot a heatmap denoting fraction of time spent in 15 by 15 view around the origin of the infinite gridworld in Figure 9. Importantly, however, the states i.e. coordinates are not the original coordinate locations of the agent but the *transformed* states (after applying symloge to the original coordinates). Thus, some original coordinates that were originally outside the 15 by 15 in Figure 8 are now within the 15 by 15 grid in Figure 9. Note that to *visualize* discrete coordinates, the transformed states were rounded to the nearest integer. In training STOP, however, there is no rounding and the true output of the state transformations is fed into the STOP agent.

Recall, that the agent observes these transformed states i.e. the transformed states are fed as input to the policy and critic. Thus, the heatmap shows the fraction of times the agent observes the *transformed* states from time 0 up till a given time-step $T$. We can see the agent never spends time outside of this transformed 15 by 15 grid i.e. the all the observed transformed states lie within this 15 by 15 view (fraction of time spent in sub-grid is 1). Moreover, upon comparing Figure 9 to Figure 8, we can see that the probability of observing a given transformed state ($\approx 0.3$) is significantly higher than observing a native state ($\approx 0.08$), indicating that from the agent's perspective, it is visiting a set of states more often.

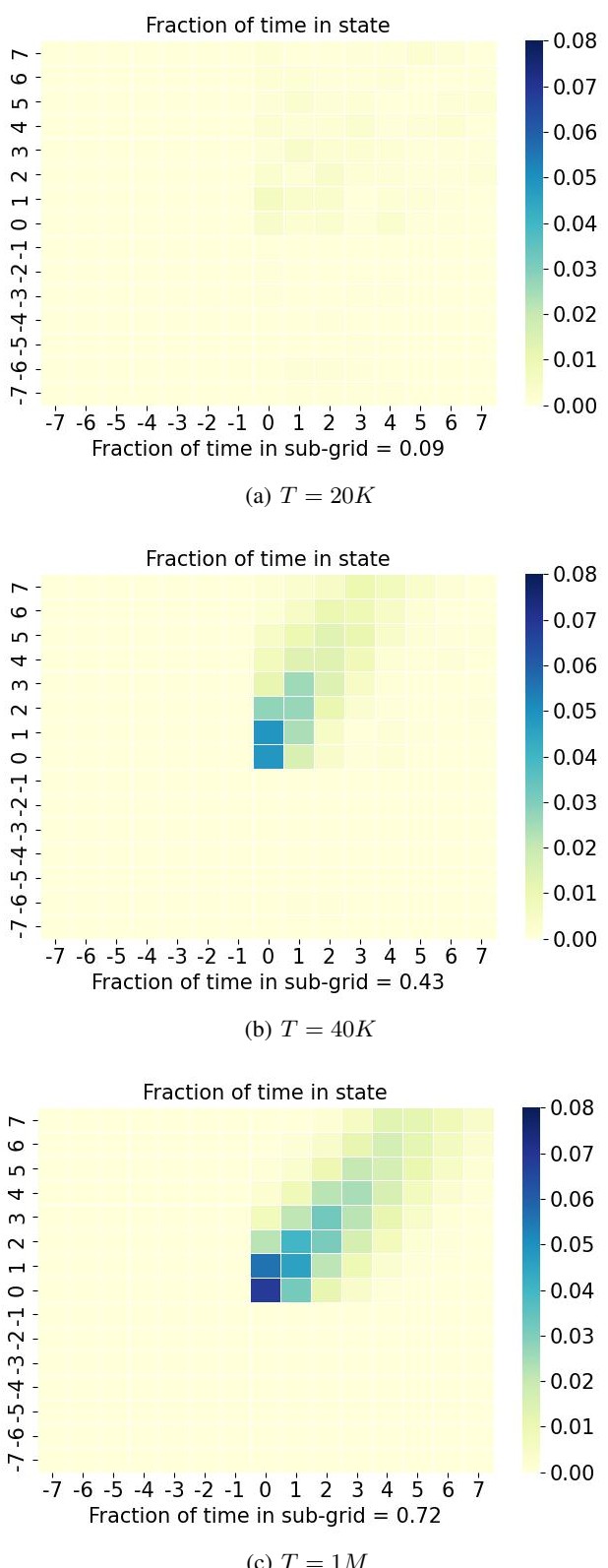

(a) $T = 20K$

(b) $T = 40K$

(c) $T = 1M$

Figure 8: Heatmap showing state visitation (as a probability of time spent in a given state) of infinite gridworld agent over time within a sub-grid. Shows fraction of time spent in each coordinate location (state) and fraction of time spent in the sub-grid. White corresponds to low and and dark blue corresponds to high visitation.

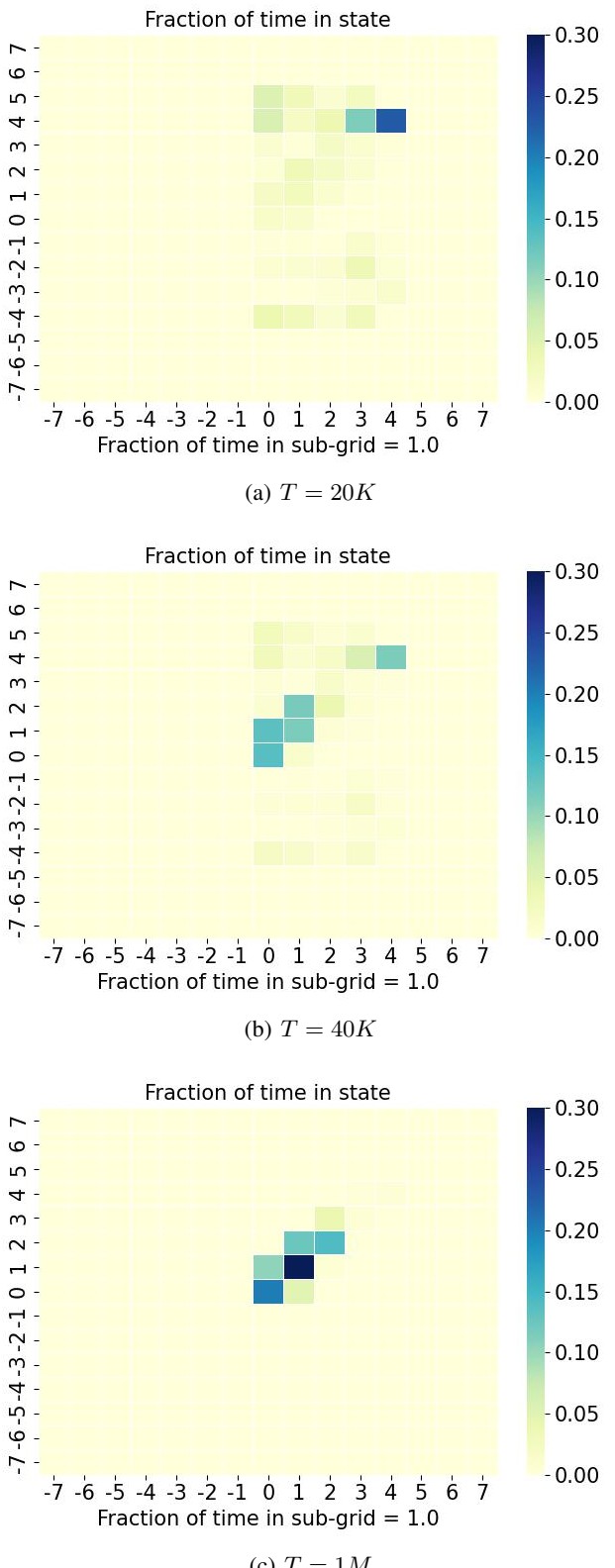

(a) $T = 20K$

(b) $T = 40K$

(c) $T = 1M$

Figure 9: Heatmap showing *transformed* state visitation (as a probability of time spent in a given state) of infinite gridworld agent over time within a sub-grid. Shows fraction of time spent in each coordinate location (transformed state) and fraction of time spent in the sub-grid. White corresponds to low and and dark blue corresponds to high visitation.

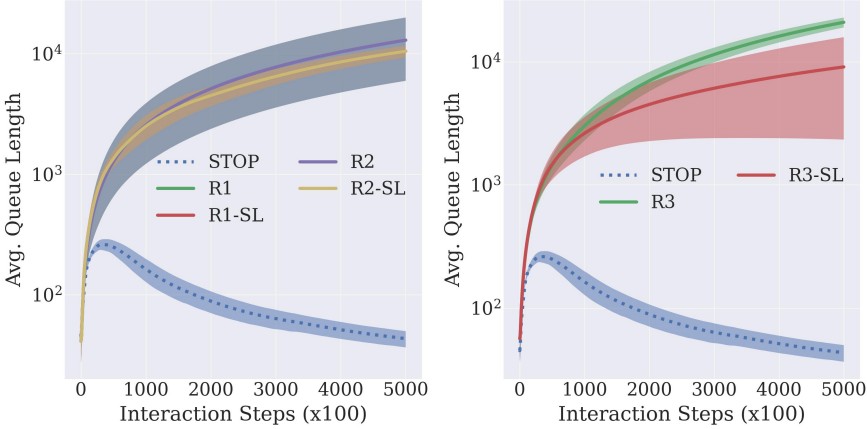

(a) Other bounded formulations of true cost  (b) Alternative optimality+stability weighting scheme

Figure 10: Average queue length on the 2-queue problem over time by different RL agents optimizing different cost formulations vs. our STOP agent. All agents diverge except for STOP. Performance metrics are computed over 5 trials with 95% confidence intervals. Lower is better.

# F   TYPICAL REWARD FUNCTION FORMULATIONS LEAD TO DIVERGENCE

In addition to those shown in Section 3, in Figure 10(a) we also include the performance of RL agents optimizing other transformations of the true optimality reward function $r(s_t)$. We consider the transformed cost: 1) $r'(s_t, a_t, s_{t+1}) := \exp(-||s_t + a_t||_2^2)$ and 2) $r'(s_t, a_t, s_{t+1}) := \exp(-||s_{t+1}||_2^2)$. The former is $R_1$ and latter is $R_2$ in the plot. We show performance when these agents do not use any state transformation and when they use the symloge transformation ($-SL$). We find that the agents still diverge, thus providing further evidence that simply transforming the true reward function in this class of problems is insufficient to yield good performance.

# G   OTHER OPTIMALITY + STABILITY COMBINATIONS

In Figure 10(b) we show that optimizing an alternative formulation of the combined optimality and stability cost: $c(s_t) + \lambda g(s_{t-1}, s_t)$ where $\lambda$ is fixed over the interaction time (no annealing) and the stability cost is treated as the regularization term is insufficient (given by $R_3$ in the plot). We tune $\lambda = \{0.2, 0.5, 1\}$ and plot the average queue length over time of the best-performing agent (lowest average queue length at the end of interaction time) for the 2-queue setting. We show performance of the best-performing agent with (-SL) and without the symloge state transformation. From Figure 10(b), we can see that these agents still diverge. Thus, further supporting our argument that the primary focus should be stability, and then optimality instead of the typical approach of focusing on optimality.

## H STOP PSEUDO-CODE

---

**Algorithm 1** STOP+PPO

---

1: Input: policy parameters $\theta_0$, critic net parameters $\phi_0$, state transformation function $\sigma$, rollout buffer $\mathcal{D}$ of length $N$.
2: **for** t = 1, 2, ... do **do**
3:     Collect sub-trajectory in rollout buffer $\{\sigma(s_k), a_k, \sigma(s_{k+1}), l_k\}_{k=1}^N$ from environment using $\pi_{\theta_{\lfloor t/N \rfloor}}$ {Note that rollout buffer contains the transformed states and the cost $l_k := \underbrace{g(s_{k-1}, s_k)}_{\text{stability cost}} + \lambda(t) \underbrace{c(s_k)}_{\text{optimality cost}}$ is a function of the non-transformed states and $\lambda$ updates according to the annealing schedule.}
4:     **if** $t \% N == 0$ **then**
5:         {Periodically update policy and critic parameters}
6:         Using rollout buffer $\mathcal{D}$ update $\theta$ and $\phi$ with average-reward PPO Zhang & Ross (2021).
7:         Empty $\mathcal{D}$
8:     **end if**
9:     Record performance of agent according to true optimality cost at time-step $t$ as a function of non-transformed state $c(s_t)$
10: **end for**

---

## I PPO WITH TRAINING WHEELS

In this section, we provide preliminary evidence that equipping PPO with training wheels i.e. a stable policy may perform worse than STOP.

In this experiment, we evaluate PPO with training wheels (PPO-TW). The PPO-TW setup closely models that of Mao et al. (2019) where we equip an on-policy policy gradient algorithm (PPO) with a stable policy. In our case, the stable policy is MAXWEIGHT (Stolyar, 2004; Tassiulas & Ephremides, 1990). MAXWEIGHT is deployed if the maximum queue length of the system exceeds 100, at which point MAXWEIGHT is used until it drives the system's maximum queue lengths to less than 50. Once it has done that, the PPO policy is deployed. Note that 1) MAXWEIGHT relies on knowing information of the transition dynamics, which can be limiting. STOP, on the other hand, does not assume access to such knowledge and 2) PPO-TW optimizes the true optimality cost (average queue length).

From Figure 11, we find that while STOP performs poorly during initial phases of learning, it is able to significantly outperform PPO-TW later on. STOP is able to learn the stabilizing and optimal actions from a destabilizing, random policy. In the case of PPO-TW, however, since the initial RL policy is poor (random) it causes the agent to diverge, which violates the safety condition often, which results in frequent deployment of the stable policy. However, this off-policy data cannot be used to update the PPO policy. Thus, the PPO policy continues to remain poor since it has inadequate data to train on, which causes the agent to diverge until it violates the safety condition, at which point the stable policy is deployed again. As noted by Mao et al. (2019), the off-policy data generated by the stable policy cannot be used to train the PPO policy. As we have noted in our future work as well, an interesting further direction will be to apply our ideas to off-policy algorithms.

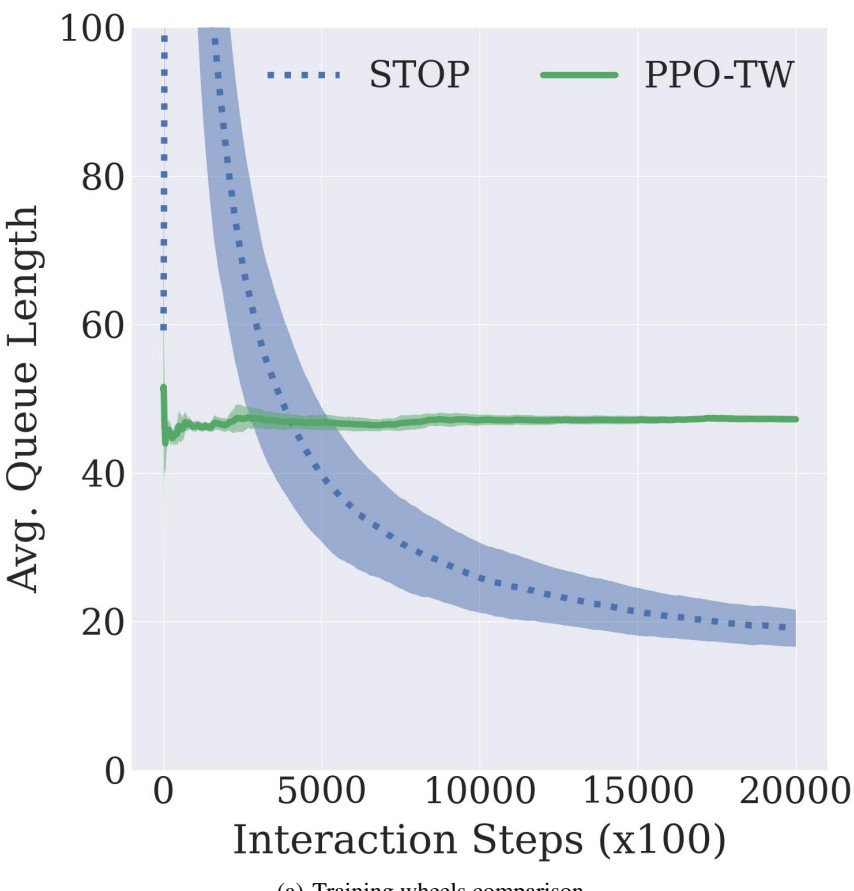

(a) Training wheels comparison

Figure 11: Average queue length on the 2-queue problem over time by our STOP agent vs. PPO-TW. Performance metrics are computed over 5 trials with 95% confidence intervals. Lower is better.

