# OpenReview forum: "Revisiting Familiar Places in an Infinite World: Continuing RL in Unbounded State Spaces"
_ICLR.cc/2024/Conference — Submitted to ICLR 2024_

### Official Review · Reviewer_bUkB · 2023-10-25

**Soundness:** 3 good
**Presentation:** 2 fair
**Contribution:** 2 fair
**Rating:** 6
**Confidence:** 3

**Summary:**

This work focuses on reinforcement learning in real-world reset-free environments where state is not necessarily bounded. It shows that vanilla policy gradient algorithms (ppo) diverge in these cases. It proposes a method, STOP, that learns to be stable at first and seek optimality afterwards, which is successful in three synthetic environments.

**Strengths:**

I think the problem that this work considers is very important, and one of the main issues that hinder the applicability of reinforcement learning in real-world systems.

**Weaknesses:**

There exists a line of work on using training wheels for RL in real systems, e.g., [Towards Safe Online Reinforcement Learning in Computer Systems](http://mlforsystems.org/assets/papers/neurips2019/towards_mao_2019.pdf), that this work ignores. Furthermore, no quantitative comparison is done with methods that learn to reset, some of which is mentioned in the related work section.

**Questions:**

1. How do you deal with exploration in the presence of stability objective? Aren't these two objectives at odds with each other?
2. How should one set the schedule for lambda? How does it interact with exploration hyper parameter schedules?
3. How can state get unbounded in a real problem? It seems to me that there is always going to be a physical limit to the state space of real systems, e.g., maximum queue lengths in the queuing environment that you explored.
4. As explained in the second paragraph of 5.2., the choice of cost function and its interplay with stability objective is problem specific. How should one choose a good cost function?
5. Are your evaluation environments based on real problems? If yes, can you elaborate on the real problems?

---

> ### Author Response · Authors · 2023-11-16
>
> Thank you to the reviewer for their comments and questions. Thank you for acknowledging that addressing these problems is important if we want to make progress towards making RL more applicable to real-world settings.
>
> **Weaknesses**
>
> **Re: cited work**
>
> Thank you for pointing out this work. We will include it in the related work sections. The work you cite is actually a deep RL version of the work by [1] that we already cite in our paper. Our comments in the related work section for [1] also apply to this new work you mention: “they assume that *a stable policy is given* and use it as a starting point for learning an optimal policy. We make no such assumption and try to learn a stable policy directly from a random policy”. We would expect methods that have access to a stable policy to perform competitively, but the main point is that we cannot always rely on access to a stable policy, and in this work, we explore the setting without access to this fixed stable policy.
>
> **Questions**
>
> **Re: exploration vs. stability**
>
> Thanks for  the question. In the class of problems we consider (see our global response), we actually want the agent to explore only in states with low norms since those correspond to low-cost states. If the agent explores in states with high norms, it incurs a high-cost penalty. STOP still encourages exploration (by inheriting PPO’s exploration strategy of sampling actions based on the logits outputted by the policy), but in low-norm/low-cost states.
>
> **Re: scheduling lambda**
>
> We set the lambda schedule as described in Section 4.2. This schedule was taken from [2]. The schedule does not interact with the exploration parameter.
>
> **Re: unbounded in real world**
>
> We understand this point. We acknowledge this in the footnote on page 1. Moreover, in the traffic control description in Section 5.1, we say: “ In this domain, SUMO models a real-life traffic situation by imposing a cap on the number of cars per lane. However, as we will show, existing deep RL algorithms still fail despite this cap”. Thus, the challenges in the unbounded state space arise even in the bounded state space setting where the bound is high.
>
> In general, a bounded state space with a high bound is not substantially different from an unbounded state space, especially in multi-dimensional problems. An unstable policy will diverge towards the boundary of the state space instead of towards infinity; see Figure 2 (d)-(f). In contrast, our approach can quickly learn to maintain the system in low-cost states and away from the boundary.
>
> **Re: problem-specific**
>
> Yes, we acknowledge this point. We do want to point out that a linear stability cost $\ell(s) = \|s\|$ is sufficient for achieving stability in most of our environments. We need to use the reciprocal stability cost only in a highly stochastic queue environment. Also, using a simple quadratic stability cost $\|s\|^2$ (inspired by domain knowledge from control theoretic and queueing problems) is sufficient in ALL our environments.
>
> We can think of adjusting the cost function as introducing domain knowledge, which is common in many RL approaches for real-world tasks. In our case, and as mentioned in Section 5.2, we made the adjustment based on relative magnitude scales between the optimality and stability cost.
>
> **Re: evaluation environments.**
>
> The experiments were conducted on simulators that share properties of the real-world task. In the appendix (section B), we include a description of the real-world setting in queueing and traffic control.
>
> [1] Bai Liu, Qiaomin Xie, and Eytan Modiano. Rl-qn: A reinforcement learning framework for optimal
> control of queueing systems
>
> [2] James MacGlashan, Evan Archer, Alisa Devlic, Takuma Seno, Craig Sherstan, Peter R. Wurman, and Peter Stone. Value function decomposition for iterative design of reinforcement learning agents

---

> > ### Author Response · Authors · 2023-11-20
> > **Following up**
> >
> > Dear reviewer, as the review discussion period is coming to an end, we kindly request you to please review the changes we have made based on your suggestions.
> >
> > Again, we would like to point out that we think the concerns are the result of minor misunderstandings that could be easily clarified (as we have tried to do in our updated pdf). We hope you will consider re-evaluating your score based on our response.
> >
> > Thank you!

---

> > > ### Comment · Reviewer_bUkB · 2023-11-21
> > > **Reviewer Response**
> > >
> > > Thank you for your detailed response.
> > >
> > > As the authors emphasized during the rebuttal, this work focuses on real-world applications.
> > >
> > > **Re: Cited Work** Does a stable policy exist in your practical application? If yes, and your focus is on the application, you should leverage that policy as the prior work did.
> > >
> > > **Re: Evaluation environments** If the specific environment and solving it is the focus of your work, the main problem and its real-world resemblance should be explained thoroughly in the body of the paper. regarding my question: I had read the explanation in the appendix about the queuing theoretic setup during the review period. However, my question was different: What is the real-world application where this setup makes sense?

---

> ### Author Response · Authors · 2023-11-22
> **Response to levereaging stable policy**
>
> Thank you for replying and your questions!
>
> **Re: cited work.**
>
> In cases where a stable policy exists, yes it can be used. However, in general, we cannot assume we have access to a stable policy. One of the main points of the paper is that while RL has been used to learn from scratch in domains like Mujoco, learning from scratch with RL in our class of domains is very challenging. We’ve shown that **learning RL policies from scratch is possible without relying on access to a stable policy** in our evaluated domains.
>
> We wish to emphasize the (somewhat nuanced) point that our work is motivated by a class of real world applications but is not itself an application paper. **Rather we see a large gap in the RL literature in that we lack RL methods that can solve this class of problems from scratch**. On the other hand, the paper you cite is an application focused paper that seeks to enable RL in the specific application of computer systems. While the load balancing example falls into the class of problems we study, it is just one instance of this problem class.
>
> Based on your suggestion, **we have included this new experiment** (see Section I in updated Appendix in main pdf) where we evaluate an *on-policy policy gradient method mixed with a stable policy as done by the authors in the paper you link* (see updated pdf for details). **We find that STOP is able to lower the queue lengths of the system much more than the RL policy with training wheels (TW; PPO-TW)**.
>
> STOP is able to learn the stabilizing and optimal actions from a destabilizing, random policy. In the case of PPO-TW, however, since the initial RL policy is poor (random) it causes the agent to diverge, which violates the safety condition often, which results in frequent deployment of the stable policy. However, this off-policy data cannot be used to update the PPO policy. Thus, the RL policy continues to remain poor since it has inadequate data to train on, which causes the agent to diverge until it violates the safety condition, at which point the stable policy is deployed again.
>
> Two remarks based on above:
> 1. In our experiment, we modeled the scenario based on the paper you linked of using an on-policy policy gradient method mixed with a stable policy. As noted in that paper and in ours, an interesting future direction is to extend our ideas to other off-policy algorithms, where we can leverage many different policies for training data.
> 2. The dependence on a stable policy is limiting due to: 1) it may not exist and 2) it may rely on access to quantities that are unknown. In the queueing experiment above, we used a stable policy called MaxWeight [4-7] which relies on knowing some information of the transition dynamics, which can be very limiting.
>
> **Re: Evaluation environments**
>
> The real-world application of this type of setup is in data-centers, wireless communication networks, and traffic intersection control [1, 2, 3]
>
>
> Thank you for the feedback! We hope we have addressed your concerns and you would consider re-evaluating your assessment.
>
> ------
> [1] R. Srikant and Lei Ying. Communication Networks: An Optimization, Control and Stochastic Networks Perspective. 2014.
>
> [2] Michael Neely. Stochastic Network Optimization with Application to Communication and Queueing Systems. 2010.
>
> [3] Ault et al. Reinforcement Learning Benchmarks for Traffic Signal Control. 2021.
>
> [4] L Tassiulas, A Ephremides, Stability properties of constrained queueing systems and scheduling policies for maximum throughput in multihop radio networks, 29th IEEE Conference on Decision and Control, 1990.
>
> [5] AL Stolyar, Maxweight scheduling in a generalized switch: State space collapse and workload minimization in heavy traffic, The Annals of Applied Probability, 2004.
>
> [6] ST Maguluri, R Srikant, Heavy traffic queue length behavior in a switch under the MaxWeight algorithm, Stochastic Systems, 2016.
>
> [7] Zixian Yang, R. Srikant, Lei Ying, Learning While Scheduling in Multi-Server Systems With Unknown Statistics: MaxWeight with Discounted UCB, Proceedings of The 26th International Conference on Artificial Intelligence and Statistics, 2023.

---

> > ### Comment · Reviewer_bUkB · 2023-11-22
> > **Reviewer Response**
> >
> > Thanks for your response and new experimental results. I increased my score.

---

> > > ### Author Response · Authors · 2023-11-22
> > >
> > > Thank you very much!

---

### Official Review · Reviewer_JKyj · 2023-10-27

**Soundness:** 2 fair
**Presentation:** 2 fair
**Contribution:** 2 fair
**Rating:** 3
**Confidence:** 4

**Summary:**

The reviewer is not an expert in average reward RL research field. This paper considers RL in average reward setting with unbounded state space and unbounded cost function. The paper first empirically demonstrated the failure or divergence encountered by existing average reward algorithm in an unbounded grid world, then proposed three techniques to stabilize learning. The techniques include a reward shaping, inspired by Lyapunov function, a scheduling process to balance the shaped and original cost, and state transformation to handle unbounded state space. The proposed method was compared with average-reward PPO in experiments, and ablation studies were conducted to compare different designs.

**Strengths:**

The originality of this paper is considering the average reward RL in an unbounded state and cost setting, which is challenging. This setting is well motivated and the authors take a lot effort to illustrate the difficulty when agent faced with unbounded state and cost.

The paper is overall easy to follow. The related works are discussed. Their own method is generally clearly described.

**Weaknesses:**

1. The authors motivate the reward shaping from the perspective of Lyapunov function, but lacks a background or detailed explanation. How is it related to analyzing the stochastic system is not presented.

2. The paper seems to emphasize the average cost may be unbounded and difficult to optimize, e.g., in Section 4.1 "As we described in Section 3, $J^O$ is difficult to minimize directly; even keeping $J^O$ bounded is challenging". However, in this setting, generally we are not directly optimizing this objective but optimizing a differential return (also mentioned in the paper). The differential value function is  bounded, and trust region or PPO style algorithm can be applied.

3. The paper lacks an algorithm description for their own method. Though several changes are described in Section 4, it is unclear to readers what value functions are used for shaped reward and original reward, how the policy is updated.

Suggestions
1. This paper considers the setting of average reward. It is better to use a subsection named "average reward MDPs" to give an overview. Though some basic terms are included in Section 2, a well organization is helpful for readers not familiar with this area.

**Questions:**

1. The unbounded grid world is unclear to me, i.e., I am unclear what is the action space of the agent. From the description in appendix and combined with arrows in Figure 1, my understanding is agent can only take actions towards the origin. For instance, in Figure 1 (a), agent can only take actions {up, left}? Please correct if I am wrong.

2. What is the intuition of Assumption 3 in Section 2? Why is divergence in the cost equivalent to the divergence in the state?

3. In section 4.3, authors mention the reward shaping and scheduling can guide agent to "re-visit regions of states with low costs". I am not very sure about this claim. Since reward shaping is dealing with a new cost $c(s_t) - c(s_{t-1})$, why learning policy on this cost can ensure agent to visit state $s$ with low cost $c(s)$?

---

> ### Author Response · Authors · 2023-11-16
>
> Thank you to the reviewer for their comments and questions. We are glad that you appreciate the effort in illustrating the difficulty of this problem. It is non-intuitive, and we wanted to stress this point.
>
> **Weaknesses**
>
> **Re: background on Lyapunov functions**
>
> We understand. We will include the necessary background in the Appendix.
>
> **Re: differential value function bounded.**
>
> Yes PPO is learning the differential value function.  However, the differential value function is well-defined only when the policy is already stable with a bounded average cost and a well defined stationary distribution. Algorithmically, average-reward PPO and TRPO methods need to first estimate the average cost before estimating the differential values; for example, see Line 4 in Algorithm 2 of https://proceedings.mlr.press/v139/zhang21q/zhang21q.pdf
>
> **Re: description of algorithm.**
>
> We have included pseudo-code in the updated appendix (section H)
>
> **Re: average reward background.**
>
> We understand. We will include the necessary background in the Appendix.
>
> **Questions**
>
> **Re: gridworld clarification**
>
> Yes, your understanding is correct. Note that in this setting, going away from the origin, even when allowed, is suboptimal at any state. It is therefore without loss of generality to only consider actions towards the origin.
>
> **Re: Assumption 3**
>
> Please see our global response.
>
> **Re: revisiting states**
>
> Please see our updated Appendix (section D) and global response.

---

> > ### Author Response · Authors · 2023-11-20
> > **Following up**
> >
> > Dear reviewer, as the review discussion period is coming to an end, we kindly request you to please review the changes we have made based on your suggestions.
> >
> > Again, we would like to point out that we think the concerns are the result of minor misunderstandings that could be easily clarified (as we have tried to do in our updated pdf). We hope you will consider re-evaluating your score based on our response.
> >
> > Thank you!

---

> > ### Comment · Reviewer_JKyj · 2023-11-20
> > **Could you intuitively explain reward shaping**
> >
> > Thanks authors for answering most of the questions.
> >
> > For your method encouraging the agent to revisit the states with low cost, providing an empirical results is great. However, could you intuitively explain the reason? In my opinion, optimizing a new cost $c(s_{t})-c(s_{t-1})$ can only encourage the agent to visit the region with low $c(s_{t})-c(s_{t-1})$ instead of low $c(s_{t})$.
> >
> > Thanks!

---

> > > ### Author Response · Authors · 2023-11-21
> > >
> > > We thank the reviewer for responding and acknowledging our response!
> > >
> > > Yes, minimizing the stability cost $c(s_t) – c(s_{t-1})$ encourages the agent to transition to states with lower norm or same norm (incurs negative or zero cost). Once the agent is stable, to ensure that the agent *actually* goes to states with low true cost $c(s_t)$, we must then include the optimality term ($c(s_t)$) which is the full version of our STOP algorithm.
> > >
> > > Thank you for your question! We hope our answers and updated pdf have resolved your concerns and you would consider re-evaluating your score.

---

> > > > ### Comment · Reviewer_JKyj · 2023-11-21
> > > > **A follow up question on gridlworld**
> > > >
> > > > Thanks for the reply.
> > > >
> > > > I still have a question on the gridworld domain. As you explained above, the action that the agent is allowed to take is only those directions towards the origin. The reward is the distance to the origin, which is also informative. Could you explain why do other methods still diverge even with this strong heuristic (here the heuristic I refer to the fact that the agent can only take actions towards the origin)?

---

> ### Author Response · Authors · 2023-11-21
> **Response to gridworld question**
>
> Thank you for the prompt reply!
>
> While there are only two actions available to the agent, the difficulty lies in poor credit assignment to these two actions due to high stochasticity, poor reward functions, and unbounded state spaces. While both actions move the agent towards the origin, the agent may not actually be able to move towards the goal since with some probability two things can happen (both of which can happen at the same time-step): 1) its movement can fail (stays in the same place) and 2) it may be pushed away from the goal. Thus, while both actions are "good" in the sense they will try to make the agent move towards the goal, the actual quality of an action is determined based on the environment stochasticity.
>
> Suppose the Manhattan distance from the origin is $c$ and is large (so the agent incurs a cost of $c$). Consider the agent is in a given fixed state and is deciding between the two actions.
>
> 1) Suppose the agent takes action $a_1$ and reduces the distance to the goal by at most 1 (since it can move only one step towards the goal and environment stochasticity can push the agent away by two steps). Thus the new cost incurred is $c'_1 = c - 1$.
>
> 2) if the agent takes $a_2$ and it increases the distance by at most 1 (for the similar reason as before),  the new cost incurred is $c'_2 = c + 1$.
>
> If $c$ is large, then the costs incurred in both these situations is $\approx c$ (since $c >> 1$). At that point, the agent finds it difficult to distinguish between the quality of these two actions since it incurs the similar cost for taking either action. The inability to distinguish between good and bad actions results in divergence. We refer you to Section 3 and Section C in the Appendix as well.
>
> Of course, if the true values for the policy were actually learned, the agent could distinguish between long-term good and bad actions. However, to get the true values the agent needs many samples of the above transition due to the high stochasticity. Our paper is showing that in our setting optimizing the optimality cost functions and unbounded state spaces (see Section 3) in the reset-free and highly stochastic setting makes convergence to the true values challenging, making it difficult for the agent to determine good/bad actions, which makes the agent diverge.
>
> To mitigate this challenge, our paper is proposing the agent first optimize a stability cost (and then optimality) instead of directly optimizing the optimality cost since the former can be more informative to the agent.
>
> Hopefully this helps clarify things!

---

> ### Comment · Reviewer_JKyj · 2023-11-22
> **After Re-evaluation**
>
> Thanks for the reply from the authors, I now have a clearer picture of the paper.
>
> I think reward shaping, as a well known technique, is known to stabilize and accelerate RL. In particular, potential-based reward shaping has some nice theoretical guarantee to make algorithm still converge to the optimal policy under infinite coverage assumption. I think the quadratic potential function given in Section 4.1 belongs to this family. So I do not regard this as something new or a contribution of the proposed method. Take the grid world domain as an example, though the space of the domain is infinite, it is not quite a big issue as I believe this domain could be solved via reward shaping if we simply do tabular learning (with a very large value table to update, then there is no approximation error of the neural net). I checked the paper, and I find authors also mention (in Appendix B.4) "optimize only the stability criterion was sufficient to stabilize the agent", which as I expected.
>
> So, I acknowledge that the authors raises a very interesting problem. However, I think the motivating example is not very convincing and the technique they propose is not very significant. As a result, I will decrease my score. Still, I encourage authors to polish the paper by considering the discussions here.

---

> ### Author Response · Authors · 2023-11-22
> **Response to relation of stability cost and potential-based shaping**
>
> Thank you for your comment.
>
> While there are parallels between our stability cost and potential-based shaping in terms of form (as we also briefly remark in the paper when we introduce the stability cost), we want to make the following remarks **on how potential-based shaping [1] is distinct from our stability cost**:
>
> 1. Stabilizing the system:
>    1. The original potential-based shaping reward does not draw any connections to the notion of *stochastic stability [2,3]* (see below for meaning). Nor are we aware of any work that does draw this connection. **Our stability cost has deep connections to the notion of stochastic stability** in that if an agent optimizes our stability cost, it will learn a long-term stabilizing behavior.
>    2. Using any potential-based reward shaping does not necessarily mean it will *stochastically stabilize* the system. While there may be many “valid” potential functions (as per the original paper), it **does not mean that those reward shaping techniques will stabilize the system**. In our work, we analyze a specific type of cost that does indeed stochastically stabilize the system.
> 2. Discounted vs. undiscounted:
>    1. Potential-based shaping’s preservation of the optimal policy was specifically for the discounted return setting (note that in the conclusion section of [1], the authors say the following about setting $\gamma=1$: “theorem may no longer guarantee optimality in this case”). We are specifically concerned with the **long-term undiscounted setting where we focus on the average-reward**.
>    2. Given that introducing a discount factor introduces an effective horizon, using a potential-based reward shaping technique will not account for the true infinite horizon/long-term behavior of the agent. **That is, techniques of accelerating convergence in the discounted case does not guarantee to achieve stochastic stability in the average reward setting.**
>    3. A discounted optimal policy may not even be stable in the average setting [4]. **This stability issue lies at the heart of stochastic network control and queueing theory, and at the heart of our contribution.**
>    4. Extending the discounted return potential-based reward shaping to the undiscounted average reward setting is an interesting future direction, but is beyond the scope of this work as it is not our focus.
>
> **Stochastic stability [2, 3] refers to keeping the Markov chain positive ergodic (or informally, the agent does not diverge).**
>
> We hope that we have explained that while the **stability cost has a simple formulation and similar formulation to potential-based shaping, it is indeed novel and significant as it sheds light on these deeper notions of stochastic stability for true long-term behavior that have largely been overlooked in the literature.**
>
> ----
> [1] Policy invariance under reward transformations: Theory and application to reward shaping. Ng et al. 1999.
>
> [2] Stable Reinforcement Learning with Unbounded State Space. Shah et al. 2020.
>
> [3] Queueing Network Controls via Deep Reinforcement Learning. Dai and Gluzman. 2021.
>
> [4] J. Michael Harrison, (1975) Dynamic Scheduling of a Multiclass Queue: Discount Optimality. Operations Research 23(2):270-282.

---

> ### Author Response · Authors · 2023-11-22
> **Response to reward shaping**
>
> Regarding the comment on simply reward shaping can "solve the gridworld domain": any type of reward shaping is not necessarily sufficient to yield good performance (as we have shown in Section 3, Appendix F and G,  and in our response to other reviewers). Our work is showing that a specific type of reward shaping that encourages the Markov chain to be positive ergodic (stochastically stable) does lead to good performance. While the stability cost appears similar to the potential-based shaping, the stability cost is indeed novel for reasons we mentioned in the above response.

---

> ### Comment · Reviewer_JKyj · 2023-11-22
> **Undiscounted case**
>
> Thanks for the reply!
>
> In my opinion, the case of $\gamma=1$ is not hard to verify. $f(s,s')=\phi(s')-\phi(s)$. Consider original $G=\frac{1}{T}\sum_{t=0}^T r_t$. The one with reward shaping is $G^\phi=\frac{1}{T}\sum_{t=0}^T(r_t + f(s_t, s_{t+1}))=\frac{1}{T}(\sum_{t=0}^T r_t + \sum_{t=0}^T\phi(s_{t+1}) - \sum_{t=0}^T \phi(s_t))=\frac{1}{T}(\sum_{t=0}^T r_t + \sum_{t=0}^T\phi(s_{t})-\phi(s_0) - \sum_{t=0}^T \phi(s_t))=\frac{1}{T}\sum_{t=0}^Tr_t - \frac{1}{T}\phi(s_0)=G - \frac{1}{T}\phi(s_0)$. This will not alter the original problem.

---

> ### Author Response · Authors · 2023-11-22
>
> Makes sense and thank you for your reply! Your verification also confirms that our approach is indeed sound.
>
> However, we would still stress our main point, and we hope you can re-assess your evaluation based on it: that while the stability  cost has relations to the potential-based shaping, **the original potential-based shaping reward does not draw any connections to the notion of stochastic stability. Nor are we aware of any work that does draw this connection. Our approach to reward shaping (our stability cost) has deep connections to the notion of stochastic stability in that if an agent optimizes our stability cost, it will learn a long-term stochastic stabilizing behavior.**

---

### Official Review · Reviewer_Nors · 2023-10-27

**Soundness:** 3 good
**Presentation:** 3 good
**Contribution:** 2 fair
**Rating:** 5
**Confidence:** 4

**Summary:**

This paper aims to deal with unbounded state spaces and unbounded cost functions in continuing RL problems. A reward shaping approach and a state transformation technique are developed to tackle the two issues, respectively. The presented method is shown to outperform baseline methods significantly in several tasks.

**Strengths:**

- The proposed method achieves much better performance than the baselines.
- Many ablation studies are shown to prove the effectiveness of each component.
- There is a good related work section to distinguish the current method from existing works.

**Weaknesses:**

My major concern is that the proposed method seems to be designed for some specific tasks with special properties, such as Assumption 3 (norm equivalence).
It is true that the state space could be unbounded since we can not control it in most cases.
However, unlike the state space, the cost function is usually manually designed.
Thus, if an unbounded reward function is bad, a natural solution would be designing a better reward function which seems to be an easier fix.
For the tasks mentioned in this work, the hardness of tasks could be mainly due to poorly designed reward functions.
For example, can you compare methods with the reward function $r(s,a,s') = \exp(-||s+a||_2^2)$ or $r(s,a,s') = \exp(-||s'||_2^2)$ in the gridworld task?

Other issues are listed below.

- The paper could be better motivated with more real-world tasks satisfying Assumption 1-3.
- A demonstration of the generalization ability of the proposed method to other tasks is missing. It will be great if authors can test their method and show its advantage in several classic control tasks, such as Catcher, Pendulum, Acrobot, Walker, Hopper, Reacher, and Swimmer mentioned in this [work](https://drive.google.com/file/d/1l2sr7HRkeaOdZNaWNjgrqBELnTF90e70/view).
- There is no direct evidence to support the claim that "introducing stability cost helps guide the agent to re-visit regions of states with low costs." Is it possible to plot visited states with different methods to show that? For example, in the gridworld task.

**Questions:**

See Weaknesses.

---

> ### Author Response · Authors · 2023-11-16
>
> Thank you to the reviewer for their comments and questions. Thank you for appreciating how we conducted our empirical analysis.
>
> **Weakness**
>
> **Re: Specific tasks**
>
> Please see our global response. A key part of our paper’s message is that such testbeds are not accurately representing challenges found in some real world RL domains.
>
> **Re: suggested reward functions.**
>
> In Section 3, we show that with such bounded reward functions the agent still diverges. In the updated Appendix (section F), we also include the performance of an RL agent using the suggested reward functions and show that the agent still diverges.
>
> Part of our message is that designing reward functions for this class of problems is difficult. Common techniques of bounding and normalizing (such as in Section 3 and the suggested reward functions in the review) are not easily applicable in this setting. Thus, we need to resort to other techniques to get RL agents to perform well in these settings.
>
> We also emphasize that in many stochastic control problems, the original cost function is tied to a specific performance metric. For example, in queueing, minimizing the average queue length is equivalent to minimizing system latency. Changing the cost function will in general change the optimal policy (i.e., the optimal policy of the new MDP no longer minimizes latency). Our reward shaping approach has the property that the optimal policy remains unchanged, since the average stability cost $\lim_{T\to\infty} (1/T) \sum_{t=1}^T \mathbb{E}[g(s_{t-1}, s_t)]$ is 0 for any stable policy (and hence for the optimal policy as well); see Proposition 2.
>
> **Re: motivation for Assumption 1-3.**
>
> Please see our global response and we will improve the motivation provided in Section 1 in the camera ready.
>
> **Re: other domains.**
>
> Please see our global response.
>
> **Re: visiting states.**
>
> Please see our global response and updated Appendix (section D)

---

> > ### Author Response · Authors · 2023-11-20
> > **Following up**
> >
> > Dear reviewer, as the review discussion period is coming to an end, we kindly request you to please review the changes we have made based on your suggestions.
> >
> > Again, we would like to point out that we think the concerns are the result of minor misunderstandings that could be easily clarified (as we have tried to do in our updated pdf). We hope you will consider re-evaluating your score based on our response.
> >
> > Thank you!

---

> > > ### Comment · Reviewer_Nors · 2023-11-20
> > > **Reply**
> > >
> > > Sorry for a late reply and thank you for including the new results.
> > >
> > > Some of my concerns are addressed and I have increased the score from 3 to 5.
> > >
> > > However, one of my concerns remains which is that the proposed method is designed for some specific tasks with special properties. I do agree that making RL work in some real-world tasks is valuable. However, the contribution will be limited if the method only works for some specific tasks.

---

> ### Author Response · Authors · 2023-11-21
> **Response to method limited to specific tasks.**
>
> Thank you for responding and updating your score! We appreciate it. We are glad you agree that our goal of trying to make RL work on real-world tasks is valuable.
>
> We would like to point out that our paper is making a similar argument to yours: that **prior, intuitive techniques and methods** (such as vanilla PPO, normalization, bounding, clipping, resetting) that are typically successful in MuJoCo and Atari may be limited to those MuJoCo and Atari tasks only, and that they **do not generalize to our class of real-world inspired RL tasks**. **This research experience taught us that finding a set of universally-successful techniques is indeed challenging, and that we may need to innovate new techniques depending on the types of problems.**

---

### Official Review · Reviewer_em6q · 2023-11-05

**Soundness:** 4 excellent
**Presentation:** 2 fair
**Contribution:** 2 fair
**Rating:** 6
**Confidence:** 4

**Summary:**

This work focuses on a setting where the state-space is unbounded and learning must be done without regular episodic resets. The authors find that the unboundedness causes the agent diverges, requires the agent to extrapolate. The unbounded cost makes estimation with high variance, causing instability in policy learning.

The authors propose to encourage the agent to re-visit the states. A lyapunov-inspired reward shapping approach and a weight annealing scheme and state transformations are proposed.

The experiment shows the propose STOP method can learn in the environment with unbounded states & no reset.7

**Strengths:**

The experiments are conducted in various environments and tasks, including a toy environment, allocation queuing and traffic control. This is a big plus to the soundness of the experiments.

I like the way how this paper is written: find a problem and then solve it.

Nice to see the code is included in the supplementary material.

**Weaknesses:**

Section 4.3 state is poorly motivated. Why the state transformations can solve the extrapolate burden? I have no idea why just scaling the input to the network and make it "feels like a visited state" can improve anything. The claim "all these functions reduce divergence rate" is not grounded.

The proposed method is relatively trivial, the stability cost function is simply the temporal difference of the cost (the reverse of a reward function) function $g = c(s_t) - c(s_{t-1})$. The connection between "using this bonus term in the reward function" and "revisiting familiar places" is unclear to me.

The theoretical results is not deep enough to motivate the paper. For example I have no idea the what the proposition 1 and 2 is doing. Yes we have a policy gradient on this $g$ function and yes now the $J^S$ is  rate stable. So what do these mean? Why proving these two propositions can prove the proposed method is better in any aspect?

The presentation of the experiment is confusing. I almost consider the SUMO experiment is omitted in the paper as the Figure 2 mixes result from 3 environments without the introduction and the discussion of the evaluation metrics. We should have a section (either in Appendix or main body and has clear annotation) to discuss the very detailed meaning of the evaluation metrics and when presenting in the experiment section we need to emphasize what metric in what environment is improved/affected by what.

**Questions:**

It will be good to unify the color scheme in Fig 2, like setting "O"'s color to green across all figures.

Eq 3 looks similar to the Lagrangian reward shaping method in Safe RL. Can we setting the cost as the primary objective and set the stability cost to be the constraint? We can write Eq 3 to be like $c(s_t)  - \lambda g(s_{t-1}, s_t)$ and use traditional RL to solve. So that we can avoid the adhoc design of the annealing scheme.

Why we use the world cost instead of reward. I guess it's because in this infinite horizon setting our goal is to reduce the "average-cost" like the Eq 1.

We are now using $g = c(s_{t-1}) - c(s_t)$. Would that possible to apply the idea of multi-step TD like $g = c(s_{t-10} - c(s_t))$ or even with the idea of $TD(\lambda)$? What would happen in this case.

The annealing is too adhoc. Can we have a mechanism to dynamically adjust the preference over cost & stability cost by observing the state uncertainty, either via a distributional Q values or via some kinds of representation learning approaches?

---

> ### Author Response · Authors · 2023-11-16
> **Part 1/2**
>
> Thank you to the reviewer for their comments and questions. We appreciate that you liked the soundness of our empirical section.
>
> **Weaknesses**
>
> **Re: visiting states and divergence comment.**
>
> The state transformations do not solve the extrapolation burden. They *only mitigate* it. For example, suppose in the 2-queue setting, we have queue lengths [100, 200] and the RL agent has learned the optimal actions for this state. Now if the agent observes queue lengths [2000, 3000], which are very far from [100, 200], it will have a difficult time determining the optimal actions since it has to perform *extreme* generalization, which neural networks are bad at performing [1]. However, if we use a symlog transformation, as done in the paper, then the agent has learned optimal actions for symlog([100, 200]) = [4.6, 5.2], and can then *locally* generalize [1] to symlog([2000, 3000]) = [6.9, 7.6], which is significantly closer to the states that the agent was trained on.
>
> Please see our updated Appendix (section D and E) and global response. In Section E, we show that when states are transformed using symlog, the agent observes the transformed states a higher fraction of the time compared to observing the original states.
>
> **Re: theoretical result**
>
> The theoretical results prove the soundness of our proposed method.
> 1. Proposition 2  shows that if an agent optimizes the stability cost, it is indeed stable i.e. it will incur bounded cost, which means it will visit states with bounded norm.
> 2. Proposition 1 shows that the stability cost can be optimized by policy gradient methods, as it has a well-defined gradient *even when the current policy is unstable*.
>
> This should be contrasted with optimizing the optimality cost alone  (without the stability cost). When initialized from a random policy (which is likely to be unstable), it will not have a finite cost or policy gradient, so the agent cannot even start running a policy gradient method.
>
> **Re: experiment section**
>
> Thanks for the suggestion. We do include the evaluation metrics and environment descriptions in the Appendix (section B), but we will improve this presentation.
>
> **Questions**
>
> **Re: color coding.**
>
> Sure, we can do this for the camera ready.
>
> **Re: stability as constraint**
>
> We agree that our approach bears similarity to a Lagrangian formulation. Empirically, we observe that using a fixed lambda, even when chosen by hyperparameter sweeping, still leads to divergence. In the updated Appendix (section G), we hyperparameter sweep over different lambda and use your proposed formulation ($c(s_t) + \lambda g(s_{t-1}, s_t)$; note that the constraint is added instead of subtracted when learning) and show that the agent diverges. We note that if we combine your technique with the bounded cost/reward suggestion from reviewer Nors, the unbounded nature of the state will make the optimality reward close to 0, which will effectively make the agent optimize the stability cost/reward only, which is essentially our STOP agent without the optimality component.
>
> Treating and updating lambda as a Lagrangian multiplier is an interesting idea. Note that unlike usual constrained problems where the objective and constraint are competing, in our problem stability (the constraint) is a necessary first step to optimality (the objective). If the current policy is not yet stable, the first term in the Lagragian formulation $c(s_t) + \lambda g(s_{t-1}, s_t)$ will be diverging and hence dominating the objective. It is therefore crucial to first achieve stability and maintain stability, as done in STOP. Also, ensuring stability requires a constraint of the form $\lim_{T\to\infty} (1/T) \sum_{t=1}^T \mathbb{E}[g(s_{t-1}, s_{t})] = 0$, which is a hard, equality, average-cost constraint, making it challenging to directly apply existing constrained RL methods.
>
> **Re: use of cost instead of reward.**
>
> As mentioned in Section 2, these terms are from the control-theoretic perspective. Given our applications (queueing, traffic control) are typically researched from the control-theoretic perspective, we stuck to that terminology.
>
> **Re: multi-step**
>
> This is an interesting suggestion, and is definitely something we will consider in future work. We expect that performance would be problem-specific, and as is typical with multi-step methods, will depend on the hyperparameter $\lambda$
>
> **Re: dynamically setting annealing schedule**
>
> This is another very interesting direction. Our main point of the work is to show that an agent should first pursue stability and then optimality. To get this point across, we resorted to a fixed schedule used by [2]. Dynamically setting the weights is in itself a difficult research problem, but we agree a starting point would be to explore methods that consider state uncertainty.

---

> > ### Author Response · Authors · 2023-11-16
> > **Part 2/2**
> >
> > [1] Francois Chollet. Deep Learning with Python. Manning Publications Co., USA, 1st edition, 2017. ISBN 1617294438
> >
> > [2] James MacGlashan, Evan Archer, Alisa Devlic, Takuma Seno, Craig Sherstan, Peter R. Wurman, and Peter Stone. Value function decomposition for iterative design of reinforcement learning agents

---

> > > ### Author Response · Authors · 2023-11-21
> > > **Following up**
> > >
> > > Dear reviewer, as the review discussion period is coming to an end, we kindly request you to please review the changes we have made based on your suggestions.
> > >
> > > Again, we would like to point out that we think the concerns are the result of minor misunderstandings that could be easily clarified (as we have tried to do in our updated pdf). We hope you will consider re-evaluating your score based on our response.
> > >
> > > Thank you!

---

> > > > ### Comment · Reviewer_em6q · 2023-11-21
> > > >
> > > > Thank you for the responses! I'll keep my score.

---

### Author Response · Authors · 2023-11-16
**Global response**

We thank the reviewers for their comments and suggestions. We appreciate that the reviewers acknowledged the merits of our empirical analysis, and the importance of tackling these types of problems as a step towards making RL more applicable to real-world settings.

We address the two common concerns in this comment and address each reviewer individually below. We include an updated Appendix in the *main file pdf (not supplementary section)* based on the suggestions. In the supplementary materials, we include a diff for clarity of the changes.

**In all cases, concerns can be addressed with simple experiments or minor changes and we hope that the reviewers will reconsider their scores.**

**Re: evidence for STOP encouraging agents to revisit states.**

This point is well-taken. We provide evidence in the Appendix (section D) of the updated paper. We show that over the agent’s interaction time, STOP does indeed encourage the agent to frequently visit a set of states that correspond to high-reward states (closer to the origin in the infinite gridworld case).

This should be contrasted with an unstable policy, which diverges in the state space and does not come back to high-reward states.

**Re: Assumption 3 and relation to practical problems**

Below we discuss the motivation for Assumption 3, and point out that it can be significantly relaxed.

We stress that Assumptions 2 and 3 are realistic as they do occur in many real-world stochastic control problems (such as queuing and traffic control), which is the primary motivation for our paper. It should be noted that while these assumptions define a class of problems that is different from problems such as MuJoCo that are more typical in RL research, they are still important since they do occur in real-world settings. Moreover, our work shows that techniques that are applied on the typical RL problems such as MuJoCo fail to generalize well to this class of problems, which further supports our case that studying these types of problems is important.

Over the course of conducting this research, we found that there was a big gap when the typically-applied successful strategies for Atari and MuJoCo (as discussed in Section 3 and our response) did not generalize to our stochastic control setting, where achieving stability is crucial and non-trivial. Our work aims to shed light on the fact that there is this class of real-world-inspired problems that appear simple, but are surprisingly challenging, and that as a community we must innovate new techniques to solve these challenging problems.

Moreover, upon inspection, one can see that:
1. The lower bound in Assumption 3 is actually only needed for the candidate Lyapunov function $\ell(s)$ used in our stability cost $g(s,s’) = \ell(s’) - \ell(s) $. As mentioned in Section 4.1, $\ell$ is an algorithmic choice and it does not have to be the original cost function. The original cost function need not satisfy Assumption 3.
2. The upper bound in Assumption 3 is NOT needed. We (unnecessarily) included this upper bound in our original paper since it is natural in queueing and traffic control problems.
3. Assumption 3 is used only in the proof of our theoretical result Proposition 2.

Intuitively, the lower bound $\ell(s) >= \|s\|$ ensures that divergence in the state s is captured by the candidate Lyapunov function $\ell$, hence controlling $\ell$ implies stability in the state space. This is all we need.

We hope that we have addressed the main concerns with our response and updated pdf, and the reviewers would consider re-evaluating their score based on these updates.

---

### Author Response · Authors · 2023-11-18
**Following up**

Dear reviewers, we kindly request you to please read our response and let us know if we have resolved your concerns or if you have any further questions before the discussion period ends.

Again, we would like to point out that we think the concerns are the result of minor misunderstandings that could be easily clarified (as we have tried to do in our updated pdf) or discussed further. We hope the reviewers would consider re-evaluating their scores based on our response.

Thank you.

---

### Author Response · Authors · 2023-11-23
**Summary of discussion**

We thank the reviewers for actively engaging with us during this discussion period and for improving the quality of our work.

As the discussion period comes to an end, we just want to briefly summarize where things are:

1. **All initial concerns and clarifications about claims in the paper were resolved with simple experiments and are all included in the updated pdf in the Appendix**. We are glad our additions were well-received by the reviewers.
2. Regarding Nors' comment: "proposed method is designed for some specific tasks with special properties". We want to emphasize again that our paper is making a similar argument to that of Nors: that **prior, intuitive techniques and methods** that are typically successful in MuJoCo and Atari may be limited to those MuJoCo and Atari tasks only, and that they **do not generalize to our class of real-world inspired RL tasks**. This research experience taught us that it is therefore important to highlight such problems to the community, and present possible techniques to tackle this challenging class of problems. [Full response](https://openreview.net/forum?id=JDp3AQ2elP&noteId=tQBJBBK5lI)
3. Regarding JKyj's comment on the significance of our stability cost: while the stability cost has relations to the potential-based shaping (and is sound as verified by JKyj for $\gamma=1$), the original potential-based shaping reward does not draw any connections to the notion of stochastic stability. **Nor are we aware of any work that does draw a connection to stochastic stability. Our approach to reward shaping (our stability cost) has deep connections to the notion of stochastic stability in that if an agent optimizes our stability cost, it will learn a long-term stochastic stabilizing behavior** (see full response for meaning of stochastic stability). [Full response](https://openreview.net/forum?id=JDp3AQ2elP&noteId=7B6g9JlFFu)

Thank you again and we appreciate your time.

---

> ### Comment · Reviewer_Nors · 2023-12-03
>
> I have read other reviews as well as authors' responses. I decided to maintain my current score.

---

> ### Comment · Reviewer_JKyj · 2023-12-04
> **Comment by reviewer JKyj**
>
> I have read other discussions. I decided to maintain my score.

---

### Meta-Review · Area_Chair_uXWn · 2023-12-05

**Metareview:**

The paper attempts to combine Lyapunov-style stability considerations with undiscounted RL in unbounded state spaces. The main technique used is related to reward shaping, coupled with a tapering of the coefficient that trades off the cost and stability components of the optimisation objective.

Strengths:
 - the paper attempts to address an important issue. We want solutions returned by RL agents to be stable and work in unbounded state spaces.

Weaknesses:
- The paper relies on non-stationary rewards tuned with a heuristic (described on page 5 of the paper). This introduces hyper-parameters that control how the reward is changes that may not easily transfer across domains.
- Theoretical analysis is insufficient. The current theory section focuses on proving things about stability. It would be good to have an explicit guarantee on the performance of the resulting policy for an infinite grid-world. Stability is important but it is not the goal of RL. The goal is to have a good (=high-reward policy).
- There is no discussion of inductive bias. Supporting infinite / unbounded state spaces using function approximation techniques designed for bounded intervals is tricky and the paper does not seem to discuss this at length.

**Justification For Why Not Higher Score:**

See weakness section in meta-review. This paper attempts to do the right thing, but it does not do it very well.

**Justification For Why Not Lower Score:**

N/A

---

### Decision · Program_Chairs · 2024-01-16

Reject